# ToRL: Topology-preserving Representation Learning Of Object Deformations From Images

## Abstract

Representation learning of object deformations from images has been a long-standing challenge in various image or video analysis tasks. Existing deep neural networks typically focus on visual features (e.g., intensity and texture), but they often fail to capture the underlying geometric and topological structures of objects. This limitation becomes especially critical in areas, such as medical imaging and 3D modeling, where maintaining the structural integrity of objects is essential for accuracy and generalization across diverse datasets. In this paper, we introduce ToRL, a novel *Topology-preserving Representation Learning* model that, for the first time, offers an explicit mechanism for modeling intricate object topology in the latent feature space. We develop a comprehensive learning framework that captures object deformations via learned transformation groups in the latent space. Each layer of our network's decoder is carefully designed with an integrated smooth composition module, ensuring that topological properties are preserved throughout the learning process. Moreover, in contrast to a few related works that rely on a reference image to predict object deformations during inference, our approach eliminates this impractical requirement. To validate ToRL's effectiveness, we conduct extensive multi-class classification experiments across a wide range of datasets, including synthetic 2D images, real 3D brain magnetic resonance imaging (MRI) scans, real 3D adrenal computed tomography (CT) shapes, and real 2D facial expression images. Experimental results demonstrate that ToRL outperforms state-of-the-art methods, setting a new way to enforce topological consistency in representation learning. Our code is available at - https://anonymous.4open.science/r/ToRL-44BF/

## 1 Introduction

Recent advances in deep learning (DL) have driven remarkable progress in large-scale image analysis tasks, such as classification (Hao et al., 2023; Vilas et al., 2024), segmentation (Ke et al., 2023; You et al., 2024), and object detection (Deng et al., 2023; Liu et al., 2023; Pu et al., 2024), often achieving near-human performance. Yet, beneath these successes lies a fundamental limitation: current models largely rely on image representations learned from intensity or textures, leading to much reduced attention to the underlying geometric structure of objects (Geirhos et al., 2018; Baker et al., 2018; Malhotra et al., 2022). This oversight poses risks to high-stakes domains, including but not limited to medical imaging, robotics, or 3D modeling, where maintaining the structural integrity of objects is critical (Malhotra et al., 2021; Linsley et al., 2017; Ullman et al., 2016). While existing deep neural networks may have access to limited geometric features in the form of local edges or orientations, they tend to miss the complete object geometry and structure. This negatively impacts their ability to generalize and perform robustly across diverse datasets and applications when studying objects with preserved topology are indispensable.

To address this problem, recent research in geometric deep learning has focused on representing and synthesizing objects with predefined geometric properties through analytic math formulations of graphs or points (Bronstein et al., 2017; Masci et al., 2016; Rematas et al., 2021). While these approaches have shown promise, they often prove impractical in real-world applications where analytic formulations are unavailable or impractical. Later, other works began leveraging DL to au-

tomatically learn geometric properties of objects directly from image data (Ouyang et al., 2015; Papandreou et al., 2015; Jack et al., 2019).

However, many of these approaches simplify interior structures, resulting in a low-level or coarse representation of complex objects. Additionally, they often lack smoothness in data representation, which is essential for accurately modeling fine geometric properties. More recent efforts (Wang & Zhang, 2022) have developed a framework that leverages DL-trained geometric features of highly-detailed object deformations from groupwise image data (Dalca et al., 2019; Ding & Niethammer, 2022; Dey et al., 2021), offering a new approach to capture intricate morphological details and internal dynamics of image objects for improved classification tasks. Despite these advances, current methods still face key challenges: (i) their performance declines when objects across different classes are non-deformable, and (ii) they rely on a template or reference image for geometric feature extraction during inference, a requirement that proves impractical in many real-world scenarios. Moreover, all aforementioned methods do not fully capture the topological structures of object deformations. They are designed to encode the image differ-

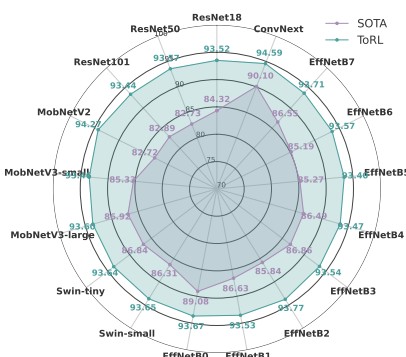

Figure 1: Classification performance validating ToRL across 17 network backbones extracting image features on Google QuickDraw. ToRL outperforms SOTA by more than 5 percent.

ences between a reference image and individual subjects in the latent feature space, which are then decoded back into the image space. The representation learning of geometric object is not explicitly modeled in the network training process. As a result, the models are incapable of accurately representing the true geometric properties, especially in scenarios where fine-grained topological understanding is crucial.

In this paper, we present ToRL, a novel topology-preserving representation learning model, that for the first time introduces an explicit modeling of intricate and complex object topology in the latent deformation space. Inspired by prior works in deformation-based representations Balakrishnan et al. (2019); Wang & Zhang (2022), our model ToRL captures object geometry down to the pixel level. Based on the premise that each object can be formulated as a deformed variant of an ideal template/reference, we incorporate proper topological constraints by regularizing the resulting deformations between the reference and each individual image. Such constraints will be carefully designed as a learning module throughout the representation learning process from groupwise images. The contributions of our proposed method are threefold:

- We develop ToRL, a new approach to model complex objects' topology via learned transformation groups in the latent space of object deformations derived from images.

- We design a novel network architecture for the decoder, incorporating an integrated smooth group composition module in the deformation space to ensure the preservation of topological properties throughout the learning process.

- In contrast to previous related works that rely on a reference image to predict object deformations during inference, our approach eliminates this impractical requirement.

We validate the effectiveness of our model in the context of binary/multi-class classification across diverse datasets, including synthetic 2D Google QuickDraw dataset (Jongejan et al., 2016), real 3D Brain MRIs (Jack Jr et al., 2008), real Adrenal CTs (Yang et al., 2023), and real 2D facial expression images (Gao et al., 2007). Experimental results show that our model achieved improved performance compared to the SOTA models, effectively preserved the topological structure of objects in images, and generalized to a wide variety of network backbones in classification tasks (see exemplary comparisons on Google Quickdraw in Fig. 1).

## 2   RELATED WORKS AND BACKGROUND

**Representation of Object Deformations.** Over the decades, significant progress has been made from traditional to DL-based representation learning of object deformations from images (Ver-

cauteren et al., 2009; Avants et al., 2008; Beg et al., 2005; Joshi et al., 2004; Wang & Zhang, 2022). With the underlying assumption that objects of a generic class can be described as deformed versions of the others, descriptors of that class arise naturally by transforming/deforming a reference image to all the other images in that class (Avants et al., 2008; Reuter et al., 2012; Joshi et al., 2004). The resulting transformation is then considered as a representation that reflects geometric object changes. In theory, every topological property of the deformed reference can be preserved by enforcing the transformation field to be diffeomorphisms, i.e., differentiable, bijective mappings with differentiable inverses (Beg et al., 2005; Arnold, 1966; Miller et al., 2006). Examples of generated images with vs. without well-preserved topology are shown in Fig. 2. Violations of topological constraints in the deformation space introduce artifacts, such as tearing, crossing, or passing through itself (see pointed arrows in the deformation fields in Fig. 2).

Given a reference image $S$ and a target image $T$ defined on a $d$-dimensional torus domain $\Omega = \mathbb{R}^d/\mathbb{Z}^d$ ($S(x), T(x) : x \in \Omega \to \mathbb{R}$), let us model the group of diffeomorphisms by a Lie group $\mathcal{G}$. A diffeomorphic transformation, $\phi_t \in \mathcal{G}$, for $t \in [0,1]$, is defined as a smooth flow over time to deform a reference image to match a target image. In this paper, we assume that both $\mathcal{G}$ and $\Omega$ are discretized and finite-dimensional. The process of deforming images $S$ by transformation $\phi_t$ is modeled by a smooth mapping

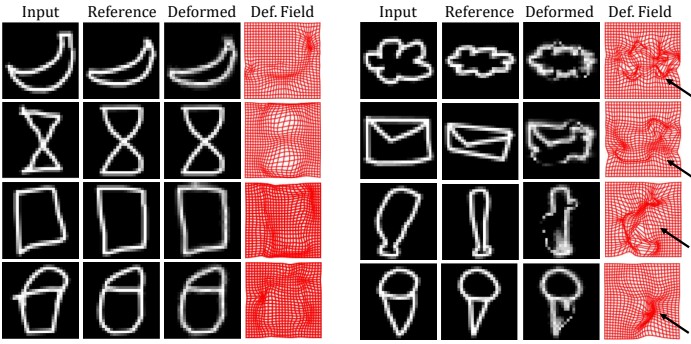

Figure 2: Examples of generated deformed images with (left panel) and without preserved topology (right panel).

$$f : \mathcal{G} \times \Omega \to \Omega, \ (\phi_t, S) \to \phi_t \cdot S.$$

Note that $\phi_t \cdot S$ is simply a notation for $f(\phi_t, S)$, and the $\cdot$ denotes a group action, i.e. the image $S$ transformed under the group action of $\phi_t$. In practice, the group action is implemented through a interpolation operator, i.e., $\phi_t \cdot S \triangleq S \circ \phi_t^{-1}$. The diffeomorphisms $\phi_t$ is typically parameterized by its linearized time-dependent velocity fields under a large diffeomorphic deformation metric mapping (Beg et al., 2005), or a stationary velocity field (SVF) that remains constant over time (Arsigny et al., 2006). While we employ SVF in this paper, our framework is easily applicable to the other.

For a stationary velocity field $v$, the diffeomorphisms, $\phi_t$, are generated as solutions to the equation:

$$\frac{d\phi_t}{dt} = v \circ \phi_t, \ \text{s.t.} \ \phi_0 = x. \tag{1}$$

The solution of Eq. 1 is identified as a group exponential map using a scaling and squaring scheme Arsigny et al. (2006). The velocity field, $v$, is often used as representations of diffeomorphisms due to its nice properties of linearity (Wang & Zhang, 2022; Arsigny et al., 2006; Mok & Chung, 2021).

**Learning geometric deformations from groupwise images.** Consider a number of $N$ images, $\{I_1, \cdots, I_N\}$ of a group of images, the problem of learning geometric deformations of each image $I_n$ is to find optimal transformations (diffeomorphisms), $\{\phi_n, \cdots, \phi_N\}$, that minimize a defined energy function

$$E(I, v_n) = \sum_{n=1}^{N} \frac{1}{\sigma^2} \text{Dist}(I \circ \phi_n^{-1}, I_n) + \|\nabla v_n\|, \ \text{s.t. Eq. 1}, \tag{2}$$

where $\sigma^2$ is a noise variance and $\circ$ denotes an interpolation operator that deforms image $I$ with an estimated transformation $\phi_n$, which is defined as a smooth flow over time to deform a template image to a target image by a composite function. The $\text{Dist}(\cdot, \cdot)$ is a distance function that measures the dissimilarity between images, i.e., sum-of-squared differences (Beg et al., 2005), normalized cross correlation (Avants et al., 2008), and mutual information (Wells III et al., 1996).

# 3 OUR METHOD: TORL

In this section, we introduce a novel topology-preserving representation learning network, ToRL. Our model highlights two key contributions: (i) an explicit mechanism to capture complex object topology by learning transformation groups in the latent space, and (ii) a newly designed decoder equipped with a smooth group composition module that carefully integrates features from skip connections at each layer, while complying with topological constraints. This is crucial, as conventional fusion of features such as addition or concatenation in the deformation space can break the smoothness of transformation fields, leading to the violation of these constraints. An overview of our proposed architecture is shown in Fig. 3.

## 3.1 NETWORK DESIGN

**Latent representation of transformation groups.** Consider a number of $C$ image classes, there exists a number of $N_c, c \in \{1, \ldots, C\}$ images, $\{I_{N_c}^c\}$, in each class. Let $\mathcal{P}_E$ represent an $L$-layer encoder network, where the output representation at each layer $l$ is given by

$$\mathbf{E}_l = g(\mathbf{K}_l * \mathbf{E}_{l-1} + \mathbf{b}_l), \quad \text{for} \quad l = 1, 2, \ldots, L,$$

where $g(\cdot)$ is a non-linear activation function, $\mathbf{K}_l$ denotes a set of learnable convolutional filters with $*$ representing the convolution operation, and $b_l$ is a bias term. Here, the $\mathbf{E}_0$ is the initial input images to the encoder. The latent representation $z$ can therefore be defined as $z = \mathbf{E}_L = f(\mathbf{K}_L * \mathbf{E}_{L-1} + \mathbf{b}_L)$, where $\mathbf{E}_L$ represents the final output of the encoder.

The goal of our training process is to learn a latent representation of transformation groups (also known as diffeomorphisms) that act on the learned latent factors. Our encoder initially extracts the latent image feature $z$, which is then passed through a fully connected network to transform it into geometric features represented in the latent velocity space, denoted as $v$. This is followed by our transformation group module (TGM), which generates the associated transformations, $\phi_l(v_l)$, at each layer. Similar to Eq. 1, we utilize a network architecture that implements the scaling and squaring scheme (Dalca et al., 2019; Dey et al., 2021) for practical implementations. The resulting output is then fed into the decoder, $\mathcal{P}_D$. It is worth noting that while we adopt SVF in this work, our approach can be easily applied to other parameterizations of diffeomorphisms, such as the large deformation diffeomorphic metric mapping framework used in (Wang & Zhang, 2022; Ding & Niethammer, 2022).

In order to enforce topological constraints, we assume that the learned latent transformation group, $\hat{\mathcal{G}}$, follows the same principles as transformations in the input data space. That is to say, for all $\hat{\phi}, \hat{\psi} \in \hat{\mathcal{G}}$, they are required to satisfy the following axioms:

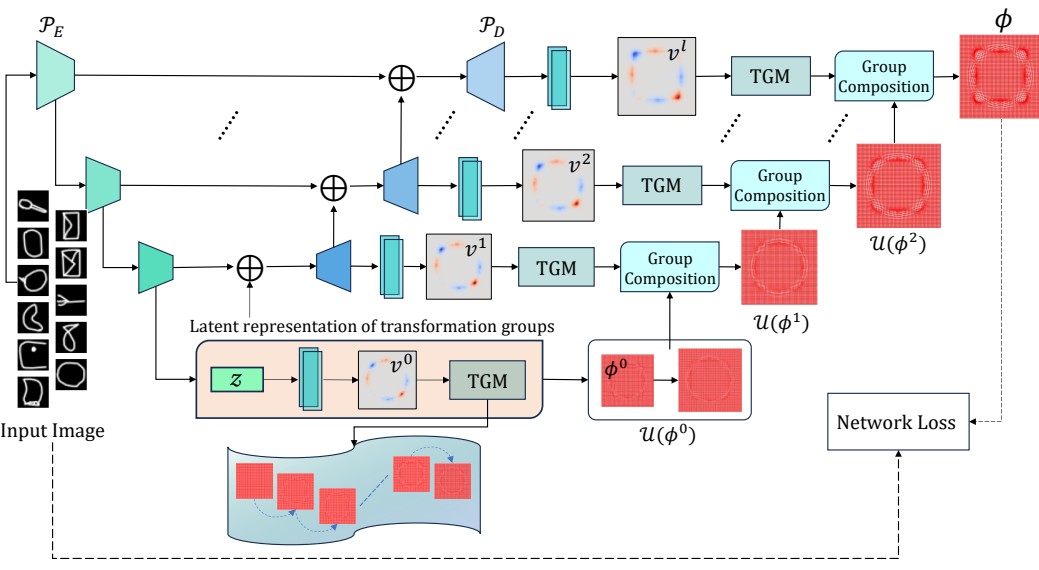

Figure 3: An overview of our proposed model ToRL.

**Axiom 1** (Closure). *The group composition of $\hat{\phi}, \hat{\psi}$ must result in an element of $\hat{\mathcal{G}}$, i.e., $\hat{\phi} \circ \hat{\psi} \in \hat{\mathcal{G}}$.*

**Axiom 2** (Associativity). *For all $\hat{\phi}, \hat{\psi}, \hat{\xi} \in \hat{\mathcal{G}}$, the group operation must be associative, i.e., $(\hat{\phi} \circ \hat{\psi}) \circ \hat{\xi} = \hat{\phi} \circ (\hat{\psi} \circ \hat{\xi})$.*

**Axiom 3** (Identity element). *There must exist an identity element $\hat{e} \in \hat{\mathcal{G}}$, such that for any transformation $\hat{\phi}$, applying the identity transformation does not alter the object, i.e., $\hat{\phi} \circ \hat{e} = \hat{e} \circ \hat{\phi} = \hat{\phi}$.*

**Axiom 4** (Inverse element). *For each transformation $\hat{\phi}$, there exists an inverse transformation $\hat{\phi}^{-1} \in \hat{\mathcal{G}}$, such that applying the transformation followed by its inverse returns to identify, i.e., $\hat{\phi} \circ \hat{\phi}^{-1} = \hat{\phi}^{-1} \circ \hat{\phi} = \hat{e}$.*

These axioms ensure that transformations in the latent space mirror the group properties in the data space, preserving structural and topological consistency across both domains. Following a similar principle, the latent transformation groups can directly act on images, $\{I\}$, at the same resolution. We require the group action to follow the rules:

$$\hat{e} \cdot I = I, \ \ \forall I \in \Omega,$$
$$\hat{\phi} \cdot (\hat{\psi} \cdot I) = (\hat{\phi}\hat{\psi}) \cdot I, \ \ \forall \hat{\phi}, \hat{\psi} \in \hat{\mathcal{G}} \text{ and } \forall I \in \Omega. \tag{3}$$

The first rule indicates that the identity transformation leaves the images unchanged. The second rule of associativity allows that a sequence of transformation groups can be composed prior to acting on the images.

**Topology-preserving decoder with group composition module.** Inspired by the U-Net architecture Ronneberger et al. (2015), we integrate skip connections into our ToRL network architecture to benefit the performance of representation learning. Specifically, we bridge higher-resolution features from the downsampling path to the corresponding layers in the upsampling path. However, previous methods Vaswani et al. (2017) that rely on simple linear addition or concatenation to merge features may fail to preserve topological constraints in the transformation fields.

To address this challenge, we introduce a novel group composition module, specifically designed to combine transformation groups in the skip connection phase. Instead of merely mixing features, our module carefully composes transformations from the upsampled layers, $\hat{\phi}_{l-1}(\hat{v}_{l-1})$, with those from the current layer, $\hat{\phi}_l(\hat{v}_l)$, ensuring the preservation of topological properties throughout the entire decoding process. Drawing on the associative rule in Eq. 3, this composition allows smooth and consistent transformations across layers.

At each layer of the decoder, $\mathcal{P}_D$, with $\mathcal{U}$ defining the upsample operator, we can formulate the composition module as follows

$$\hat{\phi}_l(\hat{v}_l) \leftarrow \hat{\phi}_l(\hat{v}_l) \circ \mathcal{U}(\hat{\phi}_{l-1}(\hat{v}_{l-1})). \tag{4}$$

Our decoder architecture (see Fig. 3) is built on the foundation described above, setting it apart from conventional approaches. Instead of using learned transpose convolutions for upsampling, we employ direct interpolation to higher-dimensional spaces. This design well maintains the smoothness and consistency of transformation grids, preserving the geometric integrity of object deformations throughout the learning process.

**Network loss.** Let $I^c$ be a reference image of class $c \in C$. For each class, there exists a set of associated deformation fields $\{\phi_1^c, \cdots, \phi_{N_c}^c\}$ between $I^c$ and each individual image $\{I_1^c, I_2^c, \ldots, I_{N_c}^c\}$. Our loss objective is to minimize

$$\mathcal{L}_{\text{ToRL}} \left( I_{N_c}^c, \mathcal{P}_D(\phi_{N_c}^c(v_{N_c}^c(z_{N_c}^c))) \cdot I^c \right).$$

Let $\Theta$ be the parameters of our ToRL architecture. Analogous to Eq. 2, we are now defining the loss function in the context of groupwise deformation representation learning for all given classes as

$$\mathcal{L}_{\text{ToRL}}(\Theta) = \sum_{c=1}^{C} \sum_{n=1}^{N_c} \frac{1}{\sigma^2} \| I_{N_c}^c - I^c \circ \phi_n^c(v_n^c(z_{N_c}^c(\Theta))) \|_2^2 + \| \nabla v_n^c(\Theta) \| + \text{reg}(\Theta), \text{ s.t. Eq. (1)}, \tag{5}$$

where $\text{reg}(.)$ is a regularity term on the network parameters. Note that in contrast to previous works (Wang & Zhang, 2022; Dalca et al., 2019; Ding & Niethammer, 2022), ToRL introduces a

fundamentally different approach: the reference image is not fed into the encoder, but is instead utilized in the loss function. Our model transforms encoded image features into the velocity space via a learned latent transformation group. This approach allows our network to leverage class-specific reference images during training and eliminate the need for reference images during testing.

## 3.2 ToRL in Downstream Task & Network optimization

We demonstrate the effectiveness of ToRL in improving performance on downstream tasks such as image classification by integrating learned representations of object deformations with image features. The flexibility of our latent representation extends beyond classification; ToRL can be integrated into a wide range of image analysis tasks, including segmentation (Ke et al., 2023; You et al., 2024), object recognition and tracking (Mao et al., 2023; Athar et al., 2023).

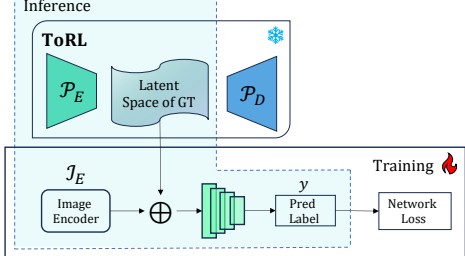

Figure 4: ToRL Classification model

Let $\mathcal{I}_E$ denote the network of an image encoder responsible for extracting image features. We integrate the latent features obtained from the ToRL model with those from the image feature extractor network to train a boosted classifier, parameterized by $\theta_c$. This classifier is designed to predict the class label $y_{nc}$ for each input image $I_{N_c}$, with a corresponding ground truth label $\hat{y}_{nc}$. While in this work we concatenate the image and shape features as $\psi(\mathcal{P}_E, \mathcal{I}_E)$, more advanced feature fusion modules can be easily integrated into this system. In this paper, we employ a cross-entropy loss for the classification loss, i.e.,

$$\mathcal{L}_{\text{clf}}(\psi(\mathcal{P}_E, \mathcal{I}_E)) = \tau \sum_{n=1}^{N_c} \sum_{c=1}^{C} -y_{nc} \cdot \log \hat{y}_{nc}(\psi(\mathcal{P}_E, \mathcal{I}_E)) + \text{reg}(\psi(\mathcal{P}_E, \mathcal{I}_E)), \qquad (6)$$

where $\tau$ is a weighting parameter.

## 4 Experiments and Evaluation

We validate the effectiveness of our model across diverse datasets, including 2D synthetic shapes, 3D real brain MRIs capturing complex neurological structures, 3D real adrenal CTs reflecting the variability and complexity of soft tissue, and 2D real facial expressions. These multi-faceted datasets covering diverse imaging modalities, dimensions, and physiological contexts underscore robustness and efficiency our model. Detailed dataset descriptions can be found in Appendix A.1

### 4.1 Experiments

We evaluate the proposed model, ToRL, from three key perspectives: (i) assessing the quality of learned latent representations by quantitatively measuring within-class and across-class feature distances in the latent space; (ii) visualizing the latent representations for 2D shapes and 3D adrenal datasets; and (iii) demonstrating its effectiveness in downstream tasks, particularly image classification. A detailed experimental evaluation plan is described as follows.

**Baseline selection.** We compare ToRL with two existing approaches for learning latent features of object deformation from groupwise images: Geo-SIC (Wang & Zhang, 2022) and CondiT (Dalca et al., 2019). These baselines have two key limitations: (i) they require a reference image during the testing phase, whereas our model ToRL does not; and (ii) they assume that objects across different classes are deformable, which restricts their application to datasets where this condition is met. To ensure a fair comparison, we select five deformable classes (circle, cloud, envelope, square, and triangle) from the Google Quickdraw dataset, following the experimental setup of Geo-SIC. For the two additional 3D datasets, since the objects are deformable across all classes, we include the entire dataset for experimental comparison.

**Evaluation of learned latent representations.** To evaluate the quality of the representations learned by ToRL, we first compare them against baseline models by leveraging these features for classifi-

cation tasks across all datasets. We train a classifier consists of three fully connected layers, with ReLU activation and a dropout layers on the learned features from all methods, and then report key performance metrics, including classification accuracy (Acc), precision (Prec), and F1-score (F1-sc). Additionally, for each testing group, we utilize a combination of inter-class divergence and intra-class compactness metrics to evaluate feature discriminability in the latent space (Li et al., 2023; Feng et al., 2024). For inter-class separability, we measure the Silhouette Score (Abavisani et al., 2020), Fisher's Discriminant Ratio (Wang et al., 2019), and KL-Divergence (Dinari & Freifeld, 2022) to show how separate the features are between classes. For intra-class compactness, we evaluate Euclidean Distance and the Davies-Bouldin Index (Abavisani et al., 2020), measuring how tightly features cluster within each class. Together, these metrics offer a comprehensive evaluation of both class separation and within-class cohesion in the latent space. To further analyze the learned latent representation of these models, we visualize the latent space using t-SNE map of all models across all datasets.

**Evaluate the benefit of ToRL in downstream tasks.** We demonstrate the effectiveness of ToRL and two baselines by comparing their learned latent representations integrated into the downstream image classification tasks. For all experiments, we use a variety of image encoders as backbones to extract latent image features, including a wide range of models such as ResNet (He et al., 2016), EfficientNet (Tan, 2019), and DenseNet (Huang et al., 2017), along with their most recent versions. To evaluate performance, we report classification accuracy (Acc), and precision (Prec).

**Evaluation of topology-preserved decoder.** To evaluate the quality of our newly designed decoder (ND), we compare it to conventional decoders (CD) used in the two baseline models (Geo-SIC/CondiT). To determine the effectiveness of the decoders, we measure whether the topology is well-preserved during the learning process. A key metric for this evaluation is the determinant of the Jacobian (DetJac), which assesses the quality of transformations and their adherence to topological constraints. For example, there is no volume change when DetJac=1, while volume shrinks when DetJac<1 and expands when DetJac>1. The value of DetJac smaller than zero indicates an artifact or singularity in the transformation field, i.e., a failure to preserve the diffeomorphic property when the effect of folding and crossing grids occurs. We also measure RMSE and SSIM scores between the source and transformed images to evaluate the quality and accuracy of the geometric transformations across all models.

**Evaluation of computational load and ToRL components.** We demonstrate the effectiveness of the transformation group module (TGM) and group composition block by conducting comparative experiments against baseline Geo-SIC/CondiT architectures which does not consists of these geometric transformation modeling components. Next, we conduct comprehensive quantitative analysis comparing parameter count, computational complexity, training/testing times per sample, and model performance across all models, to evaluate computational efficiency and performance trade-offs.

**Parameter Setting.** We set the noise variance $\sigma = 0.01$ and batch size of 128 and 16 for all 2D and 3D experiments. For 2D shape and 3D brain experiments, we split the dataset into $70\%/15\%/15\%$ for training/validation/testing. For 3D adrenal experiments, we follow the splitting settings in the original data repository (Yang et al., 2023). For network training, we utilize the cosine annealing learning rate scheduler that starts with a learning rate of $\eta = 1e^{-3}$. We train all the models with Adam optimizer, obtain the best validation performance until convergence.

## 5 RESULTS

Tab. 1 reports the classification performance based on the learned latent representations of deformations on all datasets across all methods. Our model ToRL achieves state-of-the-art results by outperforming the nearest baselines, CondiT, by 11%, 3%, 3%, 20% in classification accuracy on 2D shape, 3D brain, 3D adrenal, and 2D face datasets, respectively. This highlights the effectiveness of ToRL in learning more efficient latent representations. Intuitively, the significant performance improvement of ToRL (particularly on 2D shape and face data, performing multi-class classification) can be attributed to its elimination of the need for a reference image during the testing phase. This allows us to leverage multiple templates during training, improving the model's capacity to capture diverse intra-class variations. In contrast, other methods are constrained to using or building a single template across groups during training to ensure compatibility during testing, which limits their flexibility and effectiveness.

Table 1: Comparison of classifications using latent feature representations across all baselines.

| Models | Geo-SIC | | | CondiT | | | ToRL | | |
|---|---|---|---|---|---|---|---|---|---|
| *Metrics* | *Acc* | *Prec* | *F1* | *Acc* | *Prec* | *F1* | *Acc* | *Prec* | *F1* |
| 2D Shape | 83.38 | 83.48 | 83.10 | 87.38 | 87.52 | 87.30 | **98.58** | **98.59** | **98.58** |
| 3D Brain | 95.00 | 91.63 | 94.64 | 94.67 | 94.05 | 94.08 | **97.92** | **97.54** | **97.91** |
| 3D Adrenal | 83.22 | 82.30 | 83.00 | 83.89 | 84.67 | 81.00 | **87.25** | **84.67** | **86.00** |
| 2D Faces | 66.93 | 66.72 | 66.73 | 62.82 | 62.52 | 62.43 | **87.30** | **87.35** | **87.22** |

Table 2 presents a comparison of latent feature distances using three across-class metrics and two within-class metrics across all baselines. ToRL demonstrates consistently better performance in both categories. The higher scores on across-class metrics indicate superior class separation, while the lower within-class scores reflect more compact clustering of samples within the same class. These findings suggest that ToRL is highly effective for downstream tasks such as image classification, offering robust inter-class differentiation and strong intra-class cohesion.

Table 2: Comparison of across-class and within-class latent feature distances across all baselines.

| Datasets | 2D Shapes | | | 3D Brains | | | 3D Adrenals | | | 2D Faces | | |
|---|---|---|---|---|---|---|---|---|---|---|---|---|
| *Models* | *Geo-SIC* | *CondiT* | *ToRL* | *Geo-SIC* | *CondiT* | *ToRL* | *Geo-SIC* | *CondiT* | *ToRL* | *Geo-SIC* | *CondiT* | *ToRL* |
| **Across↑** Silhouette | 0.0901 | 0.2178 | **0.5670** | 0.0132 | 0.0017 | **0.1374** | 0.0706 | 0.0780 | **0.0793** | 0.0455 | 0.0782 | **0.2138** |
| Fisher's Disc. | 0.3577 | 1.0996 | **3.4799** | 0.0128 | 0.0111 | **0.0352** | 0.0340 | 0.0353 | **0.0369** | 0.0328 | 0.0326 | **0.9117** |
| KLD | 23.971 | 27.279 | **38.319** | 183.81 | 178.36 | **193.42** | 11.232 | 12.248 | **12.278** | 95.705 | 89.893 | **109.24** |
| **Within↓** Euclidean | 53.825 | 45.438 | **38.905** | 1083.5 | 1090.7 | **1091.9** | 496.48 | 490.255 | **487.57** | 63.735 | 61.705 | **54.053** |
| Davies-Bouldin | 3.0775 | 1.8260 | **0.3445** | 8.4165 | 6.4860 | **2.1990** | 3.0523 | 2.9481 | **2.9370** | 8.6676 | 10.083 | **1.5580** |

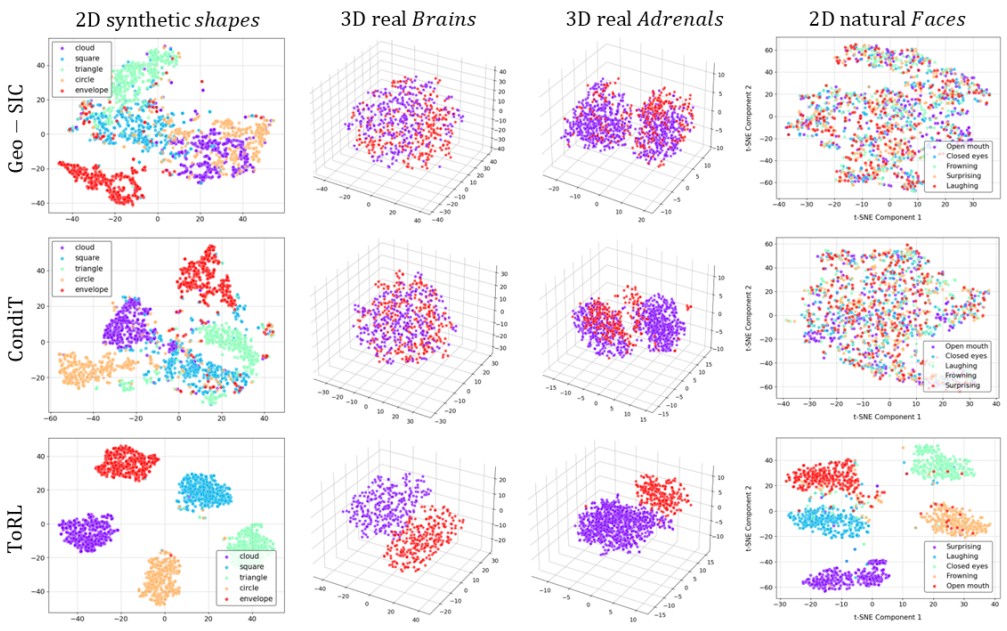

Figure 5: Latent space feature visualization using t-SNE on the 2D shapes, 3D brains, 3D adrenals, and 2D faces (from left to right) across all models. ToRL shows superior clustering in the latent spaces of object deformations.

Fig. 5 displays t-SNE visualizations of latent representations for all datasets across all models. For the 2D shape dataset, while Geo-SIC and CondiT achieve varying degrees of inter-class separation and intra-class compactness, ToRL shows clear and well-defined clusters in the latent space. In the real 3D brain and adrenal datasets, ToRL demonstrates the most distinct bimodal distribution, indicating stronger differentiation between the Normal Gland/Adrenal Mass or Healthy/Disease classes.

In contrast, Geo-SIC and CondiT show increasing levels of class overlap, suggesting limitations in their ability to learn discriminative features effectively. For the 2D natural faces dataset, ToRL exhibits superior clustering of facial expressions (surprising, laughing, frowning, open/closed mouth) compared to Geo-SIC and CondiT, which show more scattered and overlapping distributions of these emotional states. In summary, the distinct geometric patterns and clear separations visible in the latent space t-SNE plots directly correspond to ToRL's higher across-class metrics and lower within-class distances shown in the Tab. 2.

Table 3 presents classification results achieved by integrating latent representations of ToRL with image features across all datasets on three SOTA network backbones. Our model consistently outperforms the baselines, highlighting its superiority and effectiveness for downstream tasks. We present additional experiments on diverse network backbones in Appendix A.2 with an extended ablation study validating the effectiveness of ToRL and its incorporation into the downstream tasks.

Table 3: Comparison of boosted classification performance using integrated image features and latent representations from ToRL vs. other baselines.

| Backbone | Models | 2D Shapes | | 3D Brains | | 3D Adrenals | | 2D Faces | |
| | | Acc | Prec | Acc | Prec | Acc | Prec | Acc | Prec |
|---|---|---|---|---|---|---|---|---|---|
| ResNet | Geo-SIC | 90.67 | 91.27 | 94.17 | 94.83 | 85.58 | 85.16 | 74.50 | 74.29 |
| | CondiT | 88.26 | 89.01 | 95.00 | 95.44 | 84.69 | 83.67 | 70.61 | 70.40 |
| | ToRL | **99.20** | **99.19** | **97.50** | **97.52** | **87.92** | **87.80** | **93.65** | **93.54** |
| EfficientNet | Geo-SIC | 89.60 | 90.04 | 86.67 | 86.87 | 85.23 | 84.87 | 74.01 | 73.83 |
| | CondiT | 88.80 | 89.62 | 87.50 | 87.57 | 85.91 | 85.28 | 71.32 | 70.68 |
| | ToRL | **98.93** | **98.94** | **90.00** | **90.15** | **86.91** | **86.79** | **92.06** | **92.45** |
| DenseNet | Geo-SIC | 93.33 | 93.73 | 94.17 | 94.18 | 85.91 | 85.44 | 71.67 | 71.61 |
| | CondiT | 93.86 | 94.32 | 94.17 | 94.46 | 84.69 | 83.67 | 71.85 | 72.99 |
| | ToRL | **99.46** | **99.47** | **95.83** | **95.99** | **86.24** | **85.69** | **91.53** | **92.19** |

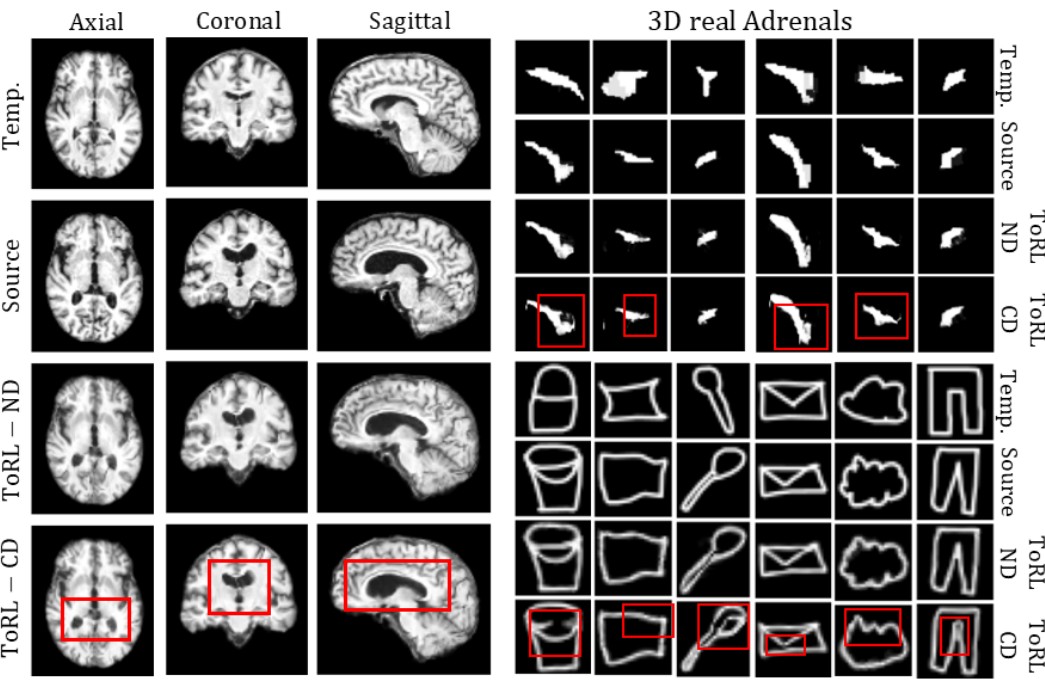

Figure 6: Comparison of the transformed images between ToRL and the baselines. Temp.: Template image, ND: New ToRL Decoder, CD: Conventional Geo-SIC/CondiT Decoder.

Fig. 6 reports a comparison between transformed images generated by ToRL with our newly designed decoder (ToRL-ND) and those generated by a conventional decoder (ToRL-CD) employed in baseline methods. ToRL-ND shows superior topological consistency and smoother deformations from the reference image to each individual image. More results can be found in Appendix. A.4.

Tab. 4 presents an evaluation of different topology preserving metrics across all models on all datasets. ToRL consistently outperforms both Geo-SiC and CondiT across all metrics, achieving the lowest RMSE, $|J_{<0}|$ and highest SSIM scores. This superior performance is particularly notable in the 3D experiments (Brains and Adrenals), where ToRL demonstrates better topology preservation as indicated by the lower DetJac values. Tab. 5 reports parameter counts, computation load (CL), training/testing times (per sample), and performances across all models on 2D and 3D datasets. ToRL emerges as a robust and efficient network architecture, achieving superior accuracy while maintaining comparatively faster inference speed as it directly predicts latent transformations without requiring template images or decoder networks during inference (unlike all the template-based baselines). Despite requiring only 10K/40K additional parameters for the TGM and group composition modules in 2D/3D experiments, respectively, ToRL delivers substantial performance improvements of $11\%/3\%$. *Please note that all the baselines require training an additional atlas building network to generate the reference image/template, which adds another level of computation load and complexities.*

Table 4: Comparison of different topology-preserving evaluation metrics across all models.

| Dat. | Model | RMSE (↓) | SSIM (↑) | ($|J_{<0}|\% \downarrow$) |
|---|---|---|---|---|
| 2D Shape | Geo-SiC | 0.1065 | 0.8605 | $1.53 \pm 0.96$ |
| | CondiT | 0.1952 | 0.6662 | $1.63 \pm 0.86$ |
| | ToRL | **0.0986** | **0.8924** | **0.72 ± 0.06** |
| 3D Brain | Geo-SiC | 0.0773 | 0.8919 | $0.03 \pm 0.03$ |
| | CondiT | 0.0755 | 0.8786 | $0.06 \pm 0.01$ |
| | ToRL | **0.0702** | **0.9047** | **0.02 ± 0.00** |
| 3D Adren. | Geo-SiC | 0.0807 | 0.9345 | $1.31 \pm 0.19$ |
| | CondiT | 0.0909 | 0.9312 | $1.34 \pm 0.18$ |
| | ToRL | **0.0595** | **0.9506** | **0.25 ± 0.06** |
| 2D Face | Geo-SiC | 0.0713 | 0.7125 | $0.64 \pm 0.05$ |
| | CondiT | 0.0980 | 0.5488 | $0.81 \pm 0.18$ |
| | ToRL | **0.0573** | **0.8835** | **0.25 ± 0.05** |

Table 5: Comparison of parameter count, computational load, and time across all models.

| Metrics | Model | Geo-SiC | CondiT | ToRL |
|---|---|---|---|---|
| Template | 2D/3D | ✓ | ✓ | × |
| TGM | 2D/3D | × | × | ✓ |
| GC | 2D/3D | × | × | ✓ |
| Params (M) | 2D | 2.14M | 2.14M | 2.15M |
| | 3D | 6.44M | 6.44M | 6.48M |
| CL (GFLOPS) | 2D | 143.03 | 143.04 | 147.02 |
| | 3D | 403.55 | 464.45 | 606.05 |
| Accuracy (%) | 2D | 83.38 | 87.52 | 98.58 |
| | 3D | 95.00 | 94.67 | 97.92 |
| Training Time | 2D | 17.61ms | 18.65ms | 32.18ms |
| | 3D | 971.3ms | 974.3ms | 1.22s |
| Testing Time | 2D | 1.14ms | 1.14ms | 1.12ms |
| | 3D | 218.3ms | 220.6ms | 178.2ms |

**Discussion.** While we need to select images that are deformable across the classes for a fair comparison with all the baselines, our model is not bound by this impractical constraint when applied to downstream tasks. To demonstrate this, we conduct an extensive analysis on the Google Quickdraw datasets, performing classification on $40$ classes. We evaluate $18$ different network backbones, representing five major families of feature extraction methods. As shown in Fig. 7, ToRL consistently outperforms all SOTA classifiers that relies on image features (Appendix A.2).

# 6 CONCLUSION

This paper presents ToRL, a novel topology-preserving representation learning model, that for the first time explicitly captures complex object topology in the latent deformation space. In contrast to existing deep neural networks that often overlook topological and geometric properties, ToRL is designed to maintain topological integrity of image objects throughout the learning process. To achieve this goal, our model directly learns transformation groups in the latent space of object deformations derived from images. The decoder architecture features a novel smooth group composition module in the deformation space, preserving topological properties during the network decoding phase. More importantly, our model ToRL eliminates the impractical reliance on a reference image for predicting the representations of object deformations during inference, which is a limitation present in current methods. Our future work includes extending ToRL to multimodal image datasets, exploring alternative transformation groups beyond stationary velocity fields, and applying it to additional downstream image analysis tasks, such as segmentation and object recognition and tracking.

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

# A  APPENDIX

## A.1  DATASET DESCRIPTIONS.

**2D Synthetic shapes.** We first randomly choose 20000 2D images from 40 distinct classes (500 images per class where the images are deformable within the class) from the Google Quickdraw data repository Jongejan et al. (2016). All images underwent affine transformation and intensity normalization with the size of $224 \times 224$.

**3D Brain MRIs.** We include 800 public T1-weighted brain MRIs from the Alzheimer's Disease Neuroimaging Initiative (ADNI) Jack Jr et al. (2008). All subjects ranged in age from 50 to 100, with 200 images each from cognitively normal (CN) and patients affected by Alzheimer's disease (AD). All MRIs were preprocessed to be the size of $104 \times 128 \times 120$, $1mm^3$ isotropic voxels, and underwent skull-stripping, intensity min-max normalization, bias-field correction, and affine registration Reuter et al. (2012)

**3D Adrenal CTs.** We select 1584 left and right real 3D adrenal glands of 792 patients from AdrenalMNIST3D data repository (Yang et al., 2023). This dataset is specifically collected to identify the presence of adrenal mass differentiating from normal adrenal glands. All images underwent affine transformation and intensity normalization with the size of $64 \times 64 \times 64$.

**2D Face Expressions.** We select 1884 real-world face images from CAS-PEAL data repository (Gao et al., 2007). Focusing on capturing different facial expressions under various background and lighting settings. We perform intensity normalization and affine transformation with the size of $128 \times 128$. Performing expression recognition tasks, we follow an identical training and testing evaluation protocol for fair comparison.

## A.2  ToR: DOWNSTREAM TASKS

Fig. 7 presents the comparative analysis between standard intensity-based SOTA networks and ToRL under various network backbones. For 2D experiments, we select different ResNet variants (ResNet18/50/101), Vision Transformers (Swin-tiny/small), ConvNext models (tiny/small), MobileNets (V2/V3), and EfficientNet series (B0-B7). For 3D experiments, we employ ResNet, DenseNet, EfficientNet, ResNext, SENet, and X3D. ToRL achieves consistent performance improvements over intensity-only models across all network backbones. While in 2D shape experiments, the baseline intensity-only networks typically achieve accuracies between $82 - 90\%$, ToRL consistently elevates performance above $90\%$, with improvements ranging from $+5.47\%$ (ConvNext-small) to $+11.55\%$ (MobNetV2). This superior performance is further evident in real 3D adrenal and brain experiments, where ToRL demonstrates significant improvements across all backbone architectures. These comprehensive experiments across diverse network backbones on both 2D and 3D datasets validate that utilizing intensity and topological features yields superior performance compared to conventional intensity-based approaches.

## A.3  ToRL: ROBUSTNESS TO INPUT PERTURBATIONS

We demonstrate the robustness of ToRL to variations in image intensity by performing a brief experiment on all real-world datasets where we add different scales of universal adversarial noises and compare ToRL with all baselines. Fig. 8 visualizes the accuracy comparison of three methods (Geo-SIC, CondiT, and ToRL) under increasing adversarial noise ($\sigma$) across three diverse real-world data, performing different binary and multiclass classification tasks. ToRL (green) consistently outperforms the other methods, maintaining higher accuracy even as noise increases from 0 to 0.05, particularly notable in facial expressions (a), brain MRIs (b), and adrenal CTs (c). All methods show performance degradation with higher noise levels, but ToRL demonstrates superior robustness.

## A.4  ABLATION STUDY: ToRL COMPONENTS

We evaluate the effectiveness of individual ToRL components through *(i) qualitative analysis of transformed images* and *(ii) quantitative assessment of boosted classification tasks*, specifically, validating the impact of the transformation group module (TGM) and group composition (GC) across a

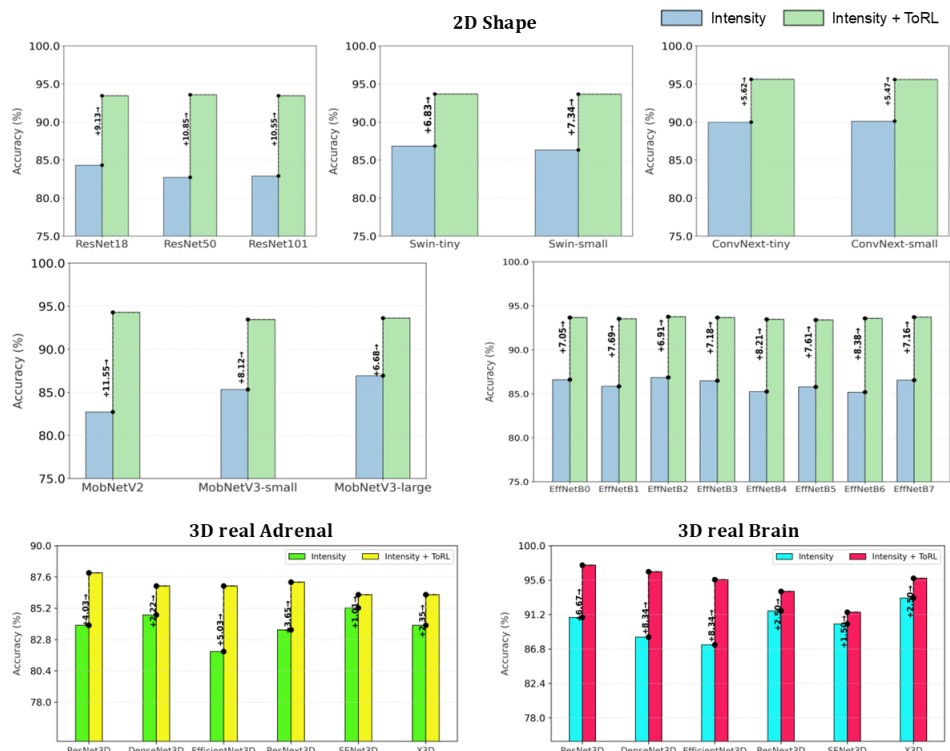

Figure 7: Effectiveness of different variants of ToRL in multi-class classification tasks under eighteen different network backbones on 2D shapes (top), 3D real adrenals (bottom left), and 3D real brains (bottom right).

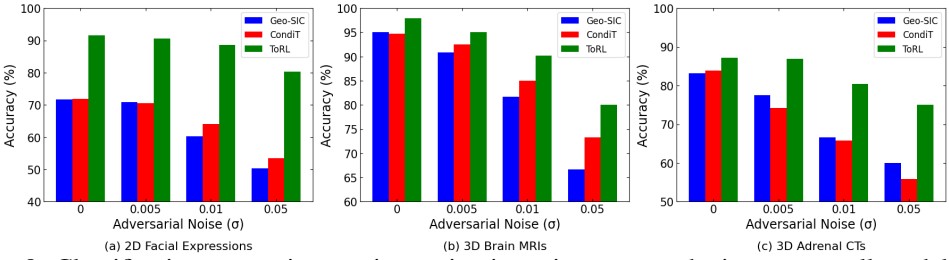

Figure 8: Classification comparison on increasing input image perturbations across all models, including ToRL.

wide variety of network backbones, ranging from lightweight networks (MobileNetV2/V3) to state-of-the-art architectures (ResNet, Swin Transformer, ConvNext, and EfficientNet variants).

Fig. 9 visualizes the transformed images across different architectural variants, where ToRL represents the complete implementation with both TGM and GC modules. The comparative analysis shows three implementations: ToRL (complete), ToRL (-GC; without Group Composition), and ToRL (-TGM, GC; without both TGM and GC modules). The red bounding boxes highlight transformation inconsistencies. These variations manifest as geometric distortions, suggesting that the absence of GC and TGM impacts transformation fidelity. The original ToRL model, having both of these components, demonstrates stable transformations, indicating that both TGM and GC components play crucial roles in maintaining structural consistency and preserving topology during the transformation process.

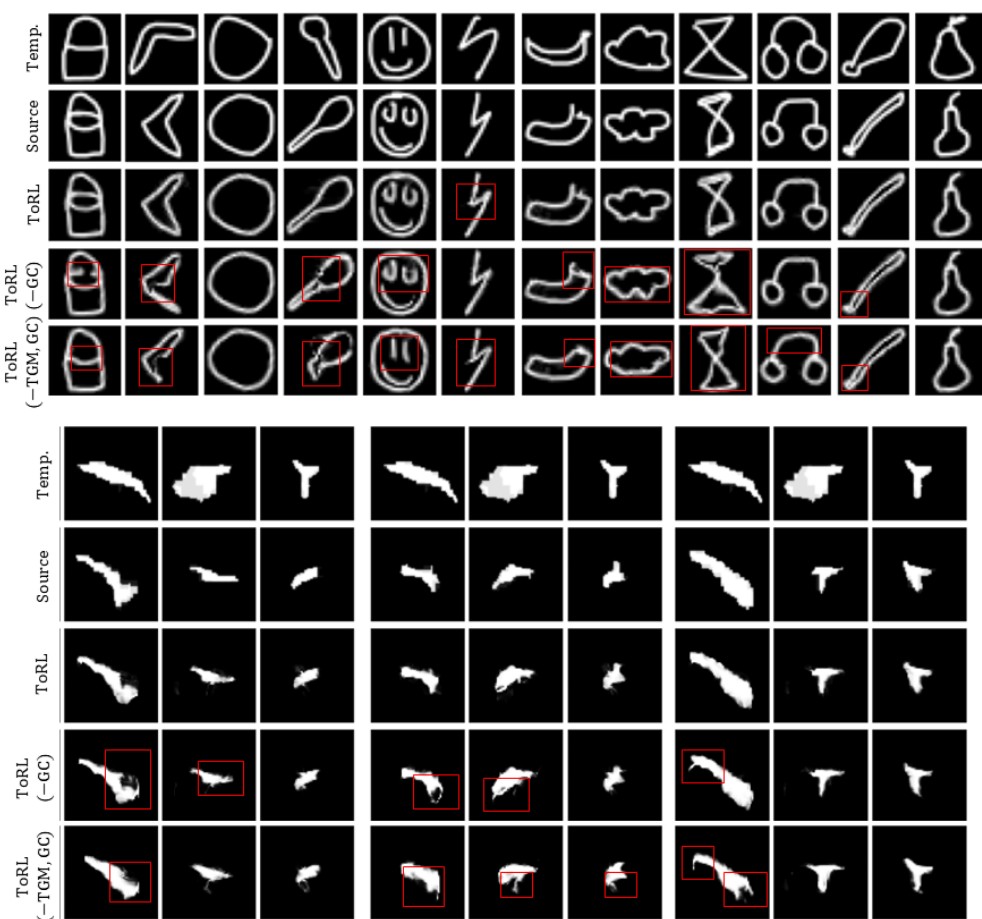

Figure 9: Ablation study on different components of ToRL based on the transformed images on 2D shapes (top) and 3D real adrenals (bottom). Temp.: Template, (-TGM, GC): without transformation group module and group composition.

Fig. 10 illustrates a boosted classification comparison between different ToRL variants with the intensity-only models, considering different network backbones. The original ToRL implementation (green), having both TGM and GC modules yields consistent $3 - 5\%$ accuracy gains under all network backbones. The ablation studies without GC (orange) and both TGM and GC (coral) show intermediate performance gains, suggesting the cumulative benefits of these modules.

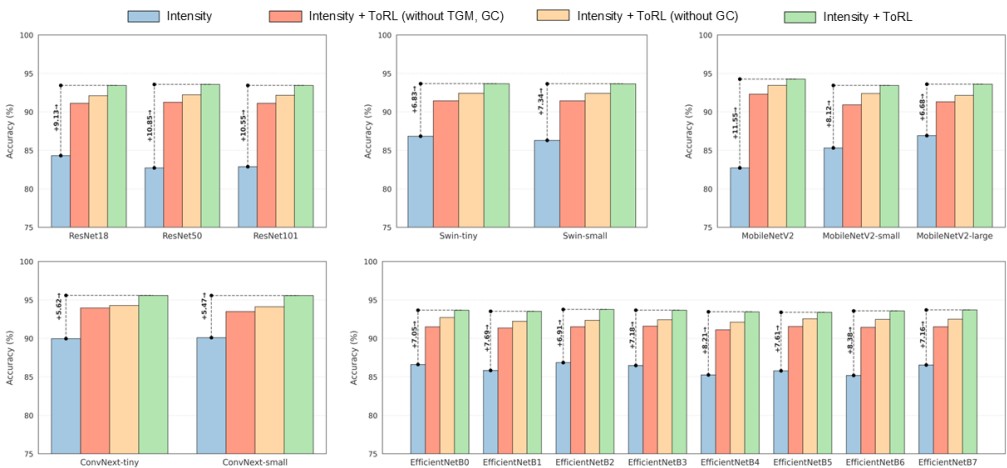

Figure 10: Classification performance comparison of different variants of ToRL.

