# OpenReview forum: "ToRL: Topology-preserving Representation Learning Of Object Deformations From Images"
_ICLR.cc/2025/Conference — Submitted to ICLR 2025_

### Official Review · Reviewer_tDX8 · 2024-10-28

**Soundness:** 3
**Presentation:** 3
**Contribution:** 3
**Rating:** 6
**Confidence:** 3

**Summary:**

Existing deep learning methods focus on visual features (like intensity or texture) but fail to preserve geometric and topological structures of objects, which is crucial for applications in medical imaging and 3D modeling. The paper introduces ToRL, a new topology-preserving representation learning framework that models complex object deformations directly from images without relying on a reference image during inference.

contributions:

1. The proposed method models intricate geometric deformations in the latent space using learned transformation groups.
This latent space representation preserves structural and topological integrity during training and decoding.

2. The decoder architecture integrates skip connections with a novel smooth group composition module. This ensures that deformations remain smooth and topologically consistent across network layers.

3. Unlike other models that require a reference image for deformation prediction, ToRL removes this dependency, improving practical usability.

**Strengths:**

1. ToRL explicitly preserves geometric and topological consistency throughout the representation learning process, which is essential for high-stakes applications such as medical imaging and 3D modeling.

2. Unlike many previous methods, ToRL eliminates the need for reference images during inference, making it more practical and efficient for real-world applications.

3. The decoder’s design with smooth transformations ensures structural continuity and smoothness, addressing the challenges of traditional feature fusion methods like simple concatenation or addition.

4. The results look good to me as it improves the classification performance and the ablation study is comprehensive.

**Weaknesses:**

1. I am concerned with the complexity. The incorporation of group transformations in the latent space and smooth group composition can increase computational overhead, making the method slower for large-scale datasets or real-time applications.

2. The custom decoder design and latent transformation modules may make it more difficult to integrate with standard architectures (like ResNet or UNet) without extensive modification.

3. Although ToRL is validated on several datasets, it might benefit from additional tests on larger or more diverse real-world datasets to further demonstrate its robustness and generalizability.

4. The strong focus on topology preservation could lead to overfitting in scenarios where topological consistency is not essential, potentially reducing the model’s flexibility.

**Questions:**

It would be great if the authors could explain more on the computational complexity of the proposed method, and evaluate on more diverse real-world datasets.

---

> ### Author Response · Authors · 2024-11-22
> **Response to Reviewer tDX8**
>
> Thank you for your positive feedback and valuable comments. Please find our point-to-point response below.
>
> > *Comment 1.* I am concerned with the complexity. The incorporation of group transformations in the latent space and smooth group composition can increase computational overhead, making the method slower for large-scale datasets or real-time applications.
>
> We conducted extensive complexity analysis of ToRL through multiple metrics, including parameter count, computational load (FLOPS), and training/testing times. Our empirical results demonstrate that the overhead is minimal - ToRL increases model size by only $0.47$% for 2D and $0.62$% for 3D experiments (additional $10$K/$40$K parameters), while delivering substantial performance gains of 11%/3% compared to baselines. This minimal parameter increase (less than $1$%) shows that the TGM and GC modules are highly parameter-efficient, making it practical for real-world applications. In fact, our ablation studies (**Tab. 5**) show that the benefits of ToRL outweigh the marginal increase in complexity, resulting in better efficiency-performance trade-off compared to simpler alternatives.
>
>
>
> > *Comment 2.* The custom decoder design and latent transformation modules may make it more difficult to integrate with standard architectures (like ResNet or UNet) without extensive modification.
>
> Please allow us to clarify that ToRL is a UNet-like architecture, and its newly designed decoder serves as a plug-and-play module that can be seamlessly incorporated into any existing UNet-like framework. To incorporate the decoder into a ResNet-like architecture, we can connect the latent features from each encoder layer to corresponding decoder layers via skip connections. The decoder’s group composition and transformation modules refine these features, ensuring multi-scale integration and spatial consistency.
>
>
>
> > *Comment 3.* Although ToRL is validated on several datasets, it might benefit from additional tests on larger or more diverse real-world datasets to further demonstrate its robustness and generalizability.
>
> Thanks for your comment. Please allow us to clarify that we performed experiments on both 2D synthetic, 3D real-world Brain and Adrenals datasets. As suggested, we performed another set of experiments, considering a real-world diverse facial expression dataset for classifying various facial expressions (**Tab. 1/2/3/4; Fig. 5**). ToRL outperformed all the baselines achieving superior performances without the impractical requirement of template images in the inference time.
>
>
> >*Comment 4.* The strong focus on topology preservation could lead to overfitting in scenarios where topological consistency is not essential, potentially reducing the model’s flexibility.
>
> We thank the reviewer for raising this concern. Please allow us to clarify that topology preservation serves as an effective regularizer that improves generalization. This approach is particularly crucial in medical applications where topological consistency enables downstream tasks like morphological analysis, anatomical measurements, and longitudinal studies. Our experiments demonstrate that preserving structural invariants enhances both classification accuracy and shape consistency, which are essential for reliable clinical quantification.
>
>
>
> In summary, we evaluated the computation complexities, discussed the generalizability of the newly designed decoder module, performed additional experiments on diverse real-world datasets, and discussed potential overfitting issues.

---

> > ### Author Response · Authors · 2024-11-25
> > **Looking forward to your post-rebuttal feedback!**
> >
> > Thank you again for the insightful comments and suggestions! Given the limited time remaining, we eagerly anticipate your subsequent feedback. It would be our pleasure to offer more responses to further demonstrate the effectiveness of our methodology.
> >
> > In our previous response, we have thoroughly reviewed your comments and provided responses summarized as follows:
> > - Reported computational complexities
> > - Explained the generalization of our proposed decoder and latent transformation group modules
> > - Experimented on an additional real-world diverse facial expression dataset
> > - Explained the overfitting issue
> >
> > We hope that the quantitative evaluation of computation costs, experiments on an additional real-world dataset, explanation regarding the generalization of the proposed decoder, and potential overfitting issues have convinced you of the merits of this paper. Again, we want to thank you for finding our technical contribution novel, experimental validation and ablation study comprehensive, and the elimination of the template image requirement applicable to real-world scenarios.
> >
> > Additionally, we wish to express our gratitude once again to you for your insightful feedback. Incorporating your suggestions has undoubtedly enhanced the clarity of our work.
> >
> > We deeply appreciate your time and effort!
> >
> > Best regards, Authors

---

> > > ### Author Response · Authors · 2024-11-26
> > > **Seeking further discussions and post-rebuttal feedback**
> > >
> > > Dear Reviewers,
> > >
> > > We would like to thank you once again for dedicating your time to review our paper.
> > >
> > > We understand that your schedule may be quite busy, especially during the holidays. As the discussion period is expected to conclude in a few days, we look forward to hearing your feedback regarding whether we have satisfactorily addressed your concerns in our explanations/clarifications and additional experiments. Please let us know if we can help you with any information/experiments/explanations. Your insights will be highly valued and deeply appreciated. Thanks.
> > >
> > >
> > > Best regards, ToRL Authors

---

> > > > ### Author Response · Authors · 2024-12-02
> > > > **Request for Final Feedback**
> > > >
> > > > Dear Reviewer tDX8,
> > > >
> > > > We would like to thank you once again for dedicating your time to review our paper. As the discussion period is expected to conclude shortly, we look forward to hearing your feedback regarding whether we have satisfactorily addressed your concerns in our Additional Clarifications. If so, we kindly request a reevaluation of the scores to positively reflect these adjustments.
> > > >
> > > > Please let us know if we can help you with any other information or queries. Your insights will be highly valued and deeply appreciated.
> > > >
> > > > Best regards, ToRL Authors

---

> > > > > ### Author Response · Authors · 2024-12-03
> > > > > **Request for considering the Rebuttal**
> > > > >
> > > > > Dear Reviewer tDX8,
> > > > >
> > > > > We sincerely appreciate your detailed feedback and the time you have invested in reviewing our work. **We have carefully addressed the points you raised and performed all additional experiments regarding your concerns.** As the discussion period is expected to conclude today, we look forward to hearing your feedback regarding whether we have satisfactorily addressed your concerns in our Additional Clarifications. If so, we kindly request a reevaluation of the scores to positively reflect these adjustments.
> > > > >
> > > > > Best Regards, ToRL Authors

---

### Official Review · Reviewer_Wezs · 2024-10-31

**Soundness:** 2
**Presentation:** 1
**Contribution:** 2
**Rating:** 5
**Confidence:** 3

**Summary:**

The paper presents a novel model and method for image representation that aims to preserve topological information. The authors highlight two main contributions: (i) developing an explicit mechanism for maintaining topological integrity within the latent space, and (ii) integrating a smooth group composition module into the skip connections of a U-shaped network to ensure topology preservation. The performance of ToRL is compared with other methods on downstream classification tasks. However, it should be noted that only classification tasks were evaluated. The main issue with this paper is that everything is artificial: the data, the tasks, and the method are all designed based on these artificial settings.

------

### Update given the author responses:
Thank you for the responses! While I still believe that the issues I highlighted — including those related to experiments, writing, and methodology — persist, I am willing to acknowledge that the quality of this work has improved. I have therefore raised my score (from 3 to 5) to reflect my perspective and leave the final decision to the AC.

**Strengths:**

1. The overall approach is somewhat inspiring and can essentially be regarded as an SVF  in the feature space.
2. The experiments demonstrate the effectiveness of the proposed representation learning method by showing improvements in downstream classification tasks

**Weaknesses:**

1. Lack of real-world applications: The data, tasks, and methods used are all based on artificial settings, similar to early spatial transformer models. Although diffeomorphism is introduced to address classification task with spatial distortion, there is no extensive exploration of SO(3) transformations or other variations, despite substantial literature in this area. There are no experiments involving real applications related to diffeomorphisms, such as registration. The classification experiments are overly simplistic, and there is no post-hoc analysis to determine if the model truly preserves topology. In fact, due to the discrete errors in SVF, it is not theoretically a true diffeomorphism, yet no analysis is provided. As a result, this paper is neither theoretical nor practical.

2. Poor writing structure and unclear contributions: The paper spends considerable space discussing diffeomorphisms, which are not directly related to its contributions, without clearly highlighting what has been contributed. The design of TGM, which could be a key part of the method, is barely mentioned. In contrast, irrelevant network architecture details with many similarities to existing models (e.g., UNet, UNet++) are overly emphasized.

3. Factual errors or lack of evidence: There is a significant factual error regarding AdrenalMNIST3D, which is a binary shape dataset, but the paper incorrectly refers to it as "3D Abdominal MRIs." If the author have visualized the data, no one will make mistakes on distinguish binary shape and MRI. In fact, even for the source, the shape is from CT scans, not MRI. This raises concerns about other potential factual errors that I may not find. Furthermore, the claim that "our approach can be easily applied to other parameterizations of diffeomorphisms" is unsupported. TGM is not adequately described, making it difficult for readers to understand its details, and it is questionable whether other NeuralODE-based methods can be seamlessly integrated into this approach.

4. Computational cost not reported or compared: The paper does not compare the parameter count and computational load against state-of-the-art methods, which may result in unfair comparisons. Additionally, the experiments do not provide comparisons of the model's parameter count and computational load (e.g., FLOPS) before and after incorporating the smooth group composition module. Therefore, it is unclear whether the observed performance improvements are due to the proposed module or simply an increase in parameters and computational cost.

**Questions:**

See weakness. Please clarify if I've misunderstood.

---

> ### Author Response · Authors · 2024-11-22
> **Response to Reviewer Wezs (I)**
>
> We appreciate your comments, questions, and feedback regarding the proposed method. Point-to-point responses are provided below.
>
> > *Comment 1.* Lack of real-world applications: The data, tasks, and methods used are all based on artificial settings, similar to early spatial transformer models. The classification experiments are overly simplistic, and there is no post-hoc analysis to determine if the model truly preserves topology.
>
> Thanks for your comment. *Please allow us to clarify that other than the experiments on 2D synthetic data, all other experiments carried out on **real-world 3D images**, performing critical clinical applications such as identifying the presence of neurodegenerative disease (Alzheimer’s) from 3D Brain MRIs and adrenal mass from 3D Adrenal CTs*. As suggested, we performed additional experiments on another diverse real-world face dataset, identifying different facial expressions (**Tab. 1/2/3/4; Fig. 5**). While all existing SOTAs require a template image to extract object deformations from images that are being used in the downstream classifier, *ToRL effectively removes this impractical requirement*, also achieving state-of-the-art performance.
>
> We would like to clarify that the focus of our paper is on improving the representation learning of object deformation from images, where topology preservation is critical. *This emphasis does not imply that our settings are artificial, as such applications are highly relevant in sensitive domains (e.g., medical and scientific visualization)*. We kindly request the reviewer to consider the revisions in our updated manuscript and re-evaluate this perspective.
>
> For post-hoc analysis of evaluating whether the model truly preserves the topology, we added a quantitative comparison of our method vs. baselines (**Tab. 4**) on the determinant of Jacobian of transformation fields. Experimental results of ToRL achieving a significantly dropped negative determinant of Jacobian indicates its superior performance in preserving the topology.
>
> > *Comment 2.* Although diffeomorphism is introduced to address classification task with spatial distortion, there is no extensive exploration of SO(3) transformations or other variations, despite substantial literature in this area.
>
> Please note that ToRL operates in the context of diffeomorphism groups, which offer significantly greater flexibility for modeling object deformations in images than the SO(3) group. This is particularly well studied in the literature of computational anatomy and shape analysis, computer graphics, robotics, and etc. More specifically, the main advantages of using diffeomorphism groups over SO(3) groups are: (i) flexibility in deformations: diffeomorphisms capture complex, nonlinear, and localized transformations, while SO(3) is limited to rigid rotations (*artificial* in real-world applications), and (ii) fine-grained topology preservation at pixel-level: diffeomorphic transformations are mathematically proven to maintain the spatial topology of images [1-3].
>
> > *Comment 3.* There are no experiments involving real applications related to diffeomorphisms, such as registration.  In fact, due to the discrete errors in SVF, it is not theoretically a true diffeomorphism, yet no analysis is provided. As a result, this paper is neither theoretical nor practical.
>
> We agree with the reviewer that in practice, the SVF may not 100% guarantee the condition of diffeomorphisms due to the numerical errors of discretization. To well examine this, we add a quantitative comparison of our method vs. baselines (**Tab. 4**) on the determinant of Jacobian of transformation fields. Experimental results of ToRL achieving a significantly dropped negative determinant of Jacobian indicates its superior performance in preserving the topology.
>
>
> > *Comment 4.* Factual errors: There is a significant factual error regarding AdrenalMNIST3D, which is a binary shape dataset, but the paper incorrectly refers to it as "3D Abdominal MRIs." If the author have visualized the data, no one will make mistakes on distinguish binary shape and MRI. In fact, even for the source, the shape is from CT scans, not MRI. This raises concerns about other potential factual errors that I may not find.
>
> We sincerely thank the reviewer for this careful observation. We apologize for the incorrect notation of AdrenalMNIST3D as MRI data. You are absolutely right - it is indeed a binary shape dataset derived from CT scans. We have corrected all instances of 'MRI' to 'CT' in our revision. To ensure complete accuracy, we have thoroughly reviewed all dataset descriptions and technical details in our paper. This was an isolated terminology error and does not affect our experimental results or conclusions, as our method was correctly applied to the CT shape data as intended.

---

> ### Author Response · Authors · 2024-11-22
> **Response to Reviewer Wezs (II)**
>
> > *Comment 5.* Poor writing structure and unclear contributions: The paper spends considerable space discussing diffeomorphisms, which are not directly related to its contributions, without clearly highlighting what has been contributed. The design of TGM, which could be a key part of the method, is barely mentioned. In contrast, irrelevant network architecture details with many similarities to existing models (e.g., UNet, UNet++) are overly emphasized.
> > *Comment 6.* Lack of evidence: Furthermore, the claim that "our approach can be easily applied to other parameterizations of diffeomorphisms" is unsupported. TGM is not adequately described, making it difficult for readers to understand its details, and it is questionable whether other NeuralODE-based methods can be seamlessly integrated into this approach.
>
>
> Please allow us to clarify that we thoroughly describe each component of ToRL such as TGM (L119-L142, L175-L235) and GC (L237-L257). Besides, we have clearly summarized our primary contributions in L80-L94. Please allow us to clarify that the backbone of our model is UNet-based architecture (similar to any deformation prediction networks [4-7]) with the incorporation of our novel transformation group module and group composition; which we thoroughly discuss in Sec. 3.1. We kindly request the reviewer to consider the network design of ToRL in Sec. 3.1.
>
> While we focus on SVF for building the transformation groups, our TGM is flexible with EPDiff or NeuralODE to develop the transformation groups.
>
>
> > *Comment 7.* Computational cost not reported or compared: The paper does not compare the parameter count and computational load against state-of-the-art methods, which may result in unfair comparisons. Additionally, the experiments do not provide comparisons of the model's parameter count and computational load (e.g., FLOPS) before and after incorporating the smooth group composition module. Therefore, it is unclear whether the observed performance improvements are due to the proposed module or simply an increase in parameters and computational cost
>
> We conducted extensive complexity analysis of ToRL through multiple metrics, including parameter count, computational load (FLOPS), and training/testing times. Our empirical results demonstrate that the overhead is minimal - ToRL increases model size by only $0.47$% for 2D and $0.62$% for 3D experiments (additional $10$K/$40$K parameters), while delivering substantial performance gains of $11$% (2D) and $3$% (3D) compared to baselines. This minimal parameter increase (less than $1$%) shows that the TGM and GC modules are highly parameter-efficient, making it practical for real-world applications. In fact, our ablation studies (**Tab. 5**) show that the benefits of ToRL outweigh the marginal increase in complexity, resulting in a better efficiency-performance trade-off compared to simpler alternatives.
>
>
> In summary, we have performed additional experiments on real-world data, assessed additional metrics to ensure topologies are being preserved, evaluated the computation complexities, and corrected the Adrenal MRIs to CTs in a revised version of the paper.

---

> > ### Author Response · Authors · 2024-11-22
> > **Response to Reviewer Wezs (III)**
> >
> > **References**
> >
> > [1] Bauer, Martin, Martins Bruveris, and Peter W. Michor. "Overview of the geometries of shape spaces and diffeomorphism groups." Journal of Mathematical Imaging and Vision 50 (2014): 60-97.
> >
> > [2] Mok, Tony CW, and Albert Chung. "Fast symmetric diffeomorphic image registration with convolutional neural networks." Proceedings of the IEEE/CVF conference on computer vision and pattern recognition. 2020.
> >
> > [3] Wyburd, Madeleine K., et al. "TEDS-Net: enforcing diffeomorphisms in spatial transformers to guarantee topology preservation in segmentations." International Conference on Medical Image Computing and Computer-Assisted Intervention. Cham: Springer International Publishing, 2021.
> >
> > [4] Wang, Jian, and Miaomiao Zhang. "Geo-sic: learning deformable geometric shapes in deep image classifiers." Advances in Neural Information Processing Systems 35 (2022): 27994-28007.
> >
> > [5] Balakrishnan, Guha, et al. "An unsupervised learning model for deformable medical image registration." Proceedings of the IEEE conference on computer vision and pattern recognition. 2018.
> >
> > [6] De Vos, Bob D., et al. "A deep learning framework for unsupervised affine and deformable image registration." Medical image analysis 52 (2019): 128-143.
> >
> > [7] Arar, Moab, et al. "Unsupervised multi-modal image registration via geometry preserving image-to-image translation." Proceedings of the IEEE/CVF conference on computer vision and pattern recognition. 2020.

---

> > > ### Author Response · Authors · 2024-11-25
> > > **Looking forward to your post-rebuttal feedback!**
> > >
> > > Thank you again for the insightful comments and suggestions! Given the limited time remaining, we eagerly anticipate your subsequent feedback. It would be our pleasure to offer more responses to further demonstrate the effectiveness of our methodology.
> > >
> > > In our previous response, we have thoroughly reviewed your comments and provided responses summarized as follows:
> > >
> > > - Performed experiments on an additional real-world facial expression dataset
> > > - Clarified the statement regarding the artificial settings
> > > - Performed post-hoc analysis to determine that ToRL efficiently preserves topological structure
> > > - Clarified the technical contributions, including explaining different ToRL components
> > > - Reported computational costs
> > > - Explained diffeomorphism groups are more flexible and precise than SO(3) for modeling deformations and preserving topology.
> > > - Solved the factual errors
> > >
> > > We hope that the quantitative evaluation on an additional real-world dataset, evaluation of different ToRL components, post-hoc analysis, computational costs comparison, and explanations regarding diffeomorphism groups, technical contribution, artificial settings have convinced you of the merits of this paper.
> > >
> > >
> > > Additionally, we wish to express our gratitude once again to you for your insightful feedback. Incorporating your suggestions has undoubtedly enhanced the clarity of our work.
> > >
> > > We deeply appreciate your time and effort!
> > >
> > > Best regards, Authors

---

> > > > ### Author Response · Authors · 2024-11-26
> > > > **Seeking further discussions and post-rebuttal feedback**
> > > >
> > > > Dear Reviewers,
> > > >
> > > > We would like to thank you once again for dedicating your time to review our paper.
> > > >
> > > > We understand that your schedule may be quite busy, especially during the holidays. As the discussion period is expected to conclude in a few days, we look forward to hearing your feedback regarding whether we have satisfactorily addressed your concerns in our explanations, clarifications, and additional ablations/experiments. Please let us know if we can help you with any information/experiments/explanations. Your insights will be highly valued and deeply appreciated. Thanks.
> > > >
> > > >
> > > > Best regards, ToRL Authors

---

> > > > > ### Comment · Reviewer_Wezs · 2024-11-26
> > > > >
> > > > > Thank you for the response and revisions. However, I must express that my concerns have not been adequately addressed, and the response has even raised new issues.
> > > > >
> > > > > On "artificial settings": Conducting experiments on real-world data does not eliminate the artificiality of the experimental setting. Running experiments on real data is a baseline expectation for AI papers, not a bonus. Even datasets like CIFAR are real-world but closer to toy datasets. Similarly, the data and settings in this paper appear toy-like. Furthermore, the purpose of the experiments themselves is questionable. Why focus on an end-to-end topological classification task when there are numerous real tasks requiring topology preservation, such as registration and image reconstruction? Classification, in the context of this paper, feels artificial.
> > > > >
> > > > > Regarding SO(3): The dismissal of existing SO(3)-invariant methods as "artificial" is difficult to accept. These foundational concepts have been integrated into Nobel-Prize achievements such as AlphaFold. The experimental and conceptual framework in this paper raises far more concerns about artificiality. Critiquing other methods requires evidence-based arguments.
> > > > >
> > > > > Writing clarity: I previously emphasized the poor writing structure, including unclear and disproportionate emphasis in the manuscript. This revision does not address those concerns, as the problematic sections remain unchanged. The authors' insistence that their writing is clear is unconvincing. Newly added content, such as the text explanation for Table 4, highlights this issue again. The description fails to clarify how the metrics were calculated or what they signify, leaving readers with more questions than answers.
> > > > >
> > > > > Significant factual error: While the authors claim to have corrected the error, references to "Adrenal CT" persist instead of the correct term, "Adrenal Shape." CT input specifically refers to data with HU values as spatial features. The original nature of the data should be made explicit so readers can fully understand the experiments.
> > > > >
> > > > > Lack of evidence: The claim that the TGM is "flexible" enough to integrate other methods remains unsupported. How? Even with differentiability, does it introduce optimization challenges or other practical barriers?
> > > > >
> > > > > I feel the authors have forced me to revisit and reread sections of the paper that were left unchanged, compelling me to reiterate points I already raised in my first-round review.

---

> ### Author Response · Authors · 2024-11-27
> **Seeking further discussions and feedbacks (I)**
>
> > On "artificial settings": Conducting experiments on real-world data does not eliminate the artificiality of the experimental setting. Running experiments on real data is a baseline expectation for AI papers, not a bonus. Even datasets like CIFAR are real-world but closer to toy datasets. Similarly, the data and settings in this paper appear toy-like.
>
> Thank you for your reply. While we respect your opinion regarding classification artificiality, we would like to emphasize that topology-preserved networks based on object deformations (specifically in image classification tasks, [1-4]) have been well studied and with the same datasets as ours [1]. While it is evident that integrating such deformable features with image intensities improved the classification performance [1] (Fig. 1/7/10, Tab. 3), such topology-preserved classification networks have difficulties in preserving object structures (Fig. 6/9). Another important drawback is such networks require template/reference images to extract those features, which is an impractical limitation considering real-life applications. To deal with these, we proposed a ToRL that efficiently preserves the object topology when predicting such important features from the latent space without requiring a pre-selected/learned template during inference.
>
> > Furthermore, the purpose of the experiments themselves is questionable. Why focus on an end-to-end topological classification task when there are numerous real tasks requiring topology preservation, such as registration and image reconstruction? Classification, in the context of this paper, feels artificial.
>
> Thanks for your comment. **Please allow us to clarify that we evaluate the quality of topology-preserved features by performing image registration (Fig. 6/9; L343-L352)**, where we found that ToRL achieves improved topological structure, having improved RMSE, SSIM, and DetJac.
>
> *Regarding classification artificiality, we want to reiterate that such classification networks have been well-studied. Our primary motivation was to efficiently preserve the topological structure and remove the burden of having template images during inference.* We kindly request the authors to reconsider this point if possible.
>
> > Regarding SO3: Regarding SO(3): The dismissal of existing SO(3)-invariant methods as "artificial" is difficult to accept. These foundational concepts have been integrated into Nobel-Prize achievements such as AlphaFold. The experimental and conceptual framework in this paper raises far more concerns about artificiality. Critiquing other methods requires evidence-based arguments.
>
> Please note that we did not say that SO(3) is not important. We only commented that considering only rigid rotations is a simplified assumption in image analysis, specifically working with medical images. The group of diffeomorphisms has more flexibility than SO(3) as diffeomorphic transformations (i) ensure smooth, invertible mappings, preserving the continuity of structures like tissues and organs without introducing folds or overlaps, which is not possible with simpler rigid or affine transformations (e.g., SO(3)), (ii) capture elastic and non-linear deformations that preserves topology and ensures realistic deformations without introducing tearing or folding of structures., which is crucial for modeling soft tissue. And these are already well-studied in the domain of deformation-based image analysis [1]. While the motivation of this paper is solely structured upon such deformation-based topology-preserved representations, we will incorporate other types of transformations in the literature study.
>
> > Writing clarity: I previously emphasized the poor writing structure, including unclear and disproportionate emphasis in the manuscript. This revision does not address those concerns, as the problematic sections remain unchanged. The authors' insistence that their writing is clear is unconvincing. Newly added content, such as the text explanation for Table 4, highlights this issue again. The description fails to clarify how the metrics were calculated or what they signify, leaving readers with more questions than answers.
>
> Please allow us to clarify that the texts above Tab. 4 discuss the experimental findings that our model preserves better topology than the baselines. Please refer to the L343-352 (blue-colored) where we thoroughly described how we calculated those matrices. We can clarify further confusion/questions if you have any regarding this.

---

> > ### Author Response · Authors · 2024-11-27
> > **Seeking further discussions and feedbacks (II)**
> >
> > > Significant factual error: While the authors claim to have corrected the error, references to "Adrenal CT" persist instead of the correct term, "Adrenal Shape." CT input specifically refers to data with HU values as spatial features. The original nature of the data should be made explicit so readers can fully understand the experiments.
> >
> > We used the same terminology (after correction) as the original paper with citations. We will extend the data description part in the Appendix of this dataset in the Appendix. Besides, we would like to request the authors if there are still "significant" factual errors in the paper.
> >
> > > Lack of evidence: The claim that the TGM is "flexible" enough to integrate other methods remains unsupported. How? Even with differentiability, does it introduce optimization challenges or other practical barriers?
> >
> > Thanks for the question. Please allow us to clarify that our newly designed decoder, having the TGM can be replaced by any vanilla decoder of UNet-like architecture. Our novel contribution is to model complex objects’ topology via learned transformation groups (TGM) in each layer of the decoder, considering negligible time and memory complexity (Tab. 5). While we focus on Stationary Velocity Fields (SVF) to learned transformation groups for the first time, one can replace this SVF module with LDDMM and other parameterizations of diffeomorphisms based on the applications. While experimenting with LDDMM and such frameworks is out of the scope of this paper, we will show a brief experimentation regarding the generalizability in the revised version.
> >
> >
> > > I feel the authors have forced me to revisit and reread sections of the paper that were left unchanged, compelling me to reiterate points I already raised in my first-round review.
> >
> > Please accept our sincere apologies. Our intention was not to force you to revisit and re-read the sections of the paper that left unchanged. We pointed out the sections based on the reviewer's raised concern regarding "... without clearly highlighting what has been contributed. The design of TGM, which could be a key part of the method, is barely mentioned". We wanted to mention that we discussed the specific contributions (TGM/GC modules and template-free topology preserved model) and thoroughly described them in the manuscript.
> >
> > As the discussion phase is still underway, we are happy to address any questions or concerns regarding our work.

---

> > > ### Author Response · Authors · 2024-11-27
> > > **Seeking further discussions and feedbacks (III)**
> > >
> > > **References**
> > >
> > > [1] Wang, Jian, and Miaomiao Zhang. "Geo-sic: learning deformable geometric shapes in deep image classifiers." Advances in Neural Information Processing Systems 35 (2022): 27994-28007.
> > >
> > > [2] Golland, Polina, et al. "Deformation analysis for shape based classification." Biennial International Conference on Information Processing in Medical Imaging. Berlin, Heidelberg: Springer Berlin Heidelberg, 2001.
> > >
> > > [3] Xing, Xiaohan, Yixuan Yuan, and Max Q-H. Meng. "Zoom in lesions for better diagnosis: Attention guided deformation network for wce image classification." IEEE Transactions on Medical Imaging 39.12 (2020): 4047-4059.
> > >
> > > [4] Chen, Zitian, et al. "Image deformation meta-networks for one-shot learning." Proceedings of the IEEE/CVF conference on computer vision and pattern recognition. 2019.
> > >
> > > [5] Yang, Jiancheng, et al. "Medmnist v2-a large-scale lightweight benchmark for 2d and 3d biomedical image classification." Scientific Data 10.1 (2023): 41.

---

### Official Review · Reviewer_W891 · 2024-11-03

**Soundness:** 3
**Presentation:** 3
**Contribution:** 2
**Rating:** 6
**Confidence:** 4

**Summary:**

This paper introduces ToRL, a novel topology-preserving representation learning model for capturing object deformations from images. Unlike existing deep neural networks that primarily focus on visual features, ToRL explicitly models object topology in the latent feature space through learned transformation groups. The model incorporates a novel decoder architecture with a smooth composition module to preserve topological properties during the learning process. Notably, ToRL eliminates the need for reference images during inference, which is a limitation in existing approaches. The authors validate their method through extensive experiments on multiple datasets, including 2D shapes and 3D medical images, demonstrating superior performance in classification tasks and better preservation of topological properties compared to state-of-the-art methods.

**Strengths:**

1.	The technical contribution is novel and well-formulated, with a clear mathematical foundation for the transformation groups and their properties in the latent space.
2.	The elimination of the reference image requirement during the inference addresses a practical limitation in existing methods, making the approach more applicable to real-world scenarios.
3.	The experimental validation is comprehensive.

**Weaknesses:**

1.	The mathematical formulation, while thorough, could benefit from more intuitive explanations or visualizations to help readers better understand the paper.
2.	The individual contributions of different components within the ToRL architecture are not analyzed. In particular, how different choices of transformation groups might affect the model's performance is not investigated.
3.	Although the paper claims superiority in preserving topological properties, additional metrics for assessing topological preservation are needed.
4.	The paper lacks a detailed analysis of the model's robustness to different types of input perturbations or variations in image quality, which is particularly relevant for medical imaging applications where image acquisition conditions can vary significantly.
5.	Beyond classification, are there any downstream tasks that could better showcase the capabilities to preserve the topology?

**Questions:**

See the weaknesses section.

---

> ### Author Response · Authors · 2024-11-22
> **Response to Reviewer W891**
>
> Thank you for the valuable comments and positive feedback on our manuscript. Please find our point-to-point responses below.
>
> > *Comment 1.* The mathematical formulation, while thorough, could benefit from more intuitive explanations or visualizations to help readers better understand the paper.
>
> Thank you for your comment. We will incorporate additional explanations for some mathematical equations (please see our response to Reviewer 2hVs for details). We kindly invite the reviewer to specify any particular formulations that require further explanation.
>
> > *Comment 2.* The individual contributions of different components within the ToRL architecture are not analyzed. In particular, how different choices of transformation groups might affect the model's performance is not investigated.
>
> Thanks for the feedback. We thoroughly investigated individual components (TGM, Group composition, and newly designed decoder) and reported all the findings in **Fig. 6/8/9** in the revised manuscript, where we found that ToRL consistently achieves superior prediction performance compared to the baselines across all datasets. We also compare ToRL with the baselines that do not have the proposed TGM and GC modules (Geo-SIC/CondiT: no TGM/GC) and evaluate different metrics, including the lowest RMSE and ratio of the negative determinant of Jacobian, where we achieve the highest structural similarity SSIM score (**Tab. 4**). These newly experimented results help highlight the contribution of different ToRL components in preserving the topology.
>
>
> > *Comment 3.* Although the paper claims superiority in preserving topological properties, additional metrics for assessing topological preservation are needed.
>
> Thanks for your feedback. In Tab. 4, we evaluate RMSE, SSIM, and ratio of the negative determinant of Jacobian to further validate the effectiveness of ToRL in topology preservation. Across all datasets, ToRL achieves the optimal performance having **$90-95$% SSIM score** with **less than $1$% negative determinant of Jacobians**. These new sets of results further validate our model’s efficiency in preserving object topology compared to the baselines.
>
>
> > *Comment 4.* The paper lacks a detailed analysis of the model's robustness to different types of input perturbations or variations in image quality, which is particularly relevant for medical imaging applications where image acquisition conditions can vary significantly.
>
> Thanks for the comments. We demonstrate the robustness of ToRL to variations in image intensity by performing a brief experiment on the real-world 2D face dataset where we add different scales of universal adversarial noises and compare ToRL with all baselines. This new set of experiments shows that ToRL consistently achieves improved performance compared to the baselines under the DenseNet backbone across different levels of adversarial attacks on image intensity. This further rectifies the efficiency of utilizing our topology-preserved learned representation in downstream classification tasks. *We included all the extended results on 3D Brain MRIs and 3D Adrenal CTs in the Appendix, Fig. 8.*
>
> | Noise-level  |    ToRL    | Geo-SIC  | CondiT  |
> |--------------|:----------:|:--------:|:-------:|
> | 0.00         | **91.53** |  71.67   |  71.85  |
> | 0.005        | **90.53** |  70.85   |  70.52  |
> | 0.01         | **88.65** |  60.32   |  64.02  |
> | 0.05         | **80.37** |  50.40   |  53.49  |
>
>
> > *Comment 5.* Beyond classification, are there any downstream tasks that could better showcase the capabilities to preserve the topology?
>
> While we primarily focus on different classification tasks (binary and multi-class) under different network backbones and diverse imaging scenarios, we also found that ToRL achieves improved performance in image alignment **(Fig. 6/8; Tab. 4)**. ToRL is flexible to be incorporated with other image analysis tasks, such as segmentation or reconstruction. We will incorporate a comprehensive evaluation of other tasks into an extended journal.

---

> > ### Author Response · Authors · 2024-11-25
> > **Looking forward to your post-rebuttal feedback!**
> >
> > Thank you again for the insightful comments and suggestions! Given the limited time remaining, we eagerly anticipate your subsequent feedback. It would be our pleasure to offer more responses to further demonstrate the effectiveness of our methodology.
> >
> > In our previous response, we have thoroughly reviewed your comments and provided responses summarized as follows:
> >
> > - Validated the different components within the ToRL architecture.
> > - Measured additional metrics, including RMSE, SSIM, and DetJac regarding the necessity of topological preservation.
> > - Presented a brief analysis other the model's robustness to input perturbation.
> > - Reported an additional downstream task and other possible extensions of our work.
> >
> >
> > We hope that the additional quantitative evaluation on different ToRL components, assessment of evaluation metrics, brief experimentation on input perturbation, and downstream tasks explanation have convinced you of the merits of this paper. Please feel free to let us know if there are additional questions. *Again, we want to thank you for finding our technical contribution novel and well-formulated, experimental validation comprehensive, and the elimination of the template image requirement applicable to real-world scenarios.*
> >
> > Additionally, we wish to express our gratitude once again to you for your insightful feedback. Incorporating your suggestions has undoubtedly enhanced the clarity of our work.
> >
> > We deeply appreciate your time and effort!
> >
> > Best regards, Authors

---

> > > ### Author Response · Authors · 2024-11-26
> > > **Seeking further discussions and post-rebuttal feedback**
> > >
> > > Dear Reviewers,
> > >
> > > We would like to thank you once again for dedicating your time to review our paper.
> > >
> > > We understand that your schedule may be quite busy, especially during the holidays. As the discussion period is expected to conclude in a few days, we look forward to hearing your feedback regarding whether we have satisfactorily addressed your concerns in our Additional Ablations and Experiments. Please let us know if we can help you with any information/experiments/explanations. Your insights will be highly valued and deeply appreciated. Thanks.
> > >
> > >
> > > Best regards, ToRL Authors

---

> > > > ### Author Response · Authors · 2024-12-02
> > > > **Request for Final Feedback**
> > > >
> > > > Dear Reviewer W891,
> > > >
> > > > We would like to thank you once again for dedicating your time to review our paper. As the discussion period is expected to conclude shortly, we look forward to hearing your feedback regarding whether we have satisfactorily addressed your concerns in our Additional Clarifications. If so, we kindly request a reevaluation of the scores to positively reflect these adjustments.
> > > >
> > > > Please let us know if we can help you with any other information or queries. Your insights will be highly valued and deeply appreciated.
> > > >
> > > > Best regards, ToRL Authors

---

> > > > > ### Comment · Reviewer_W891 · 2024-12-03
> > > > > **Thank you for your response**
> > > > >
> > > > > Thank you for your response and clarification. The response addresses most of my concerns. I am updating my score to reflect this.

---

> > > > > > ### Author Response · Authors · 2024-12-03
> > > > > > **Thanks for increasing the Score**
> > > > > >
> > > > > > Dear Reviewer W891
> > > > > >
> > > > > > We are pleased to hear that our response and clarifications effectively addressed most of your concerns. We sincerely appreciate your thoughtful consideration in updating the score from *5: marginally below the acceptance threshold* to **6: marginally above the acceptance threshold**. Please let us know if there is any additional information we can provide about our paper to address your questions and further enhance its standing.
> > > > > >
> > > > > > Best Authors, ToRL Authors

---

### Official Review · Reviewer_2hVs · 2024-11-04

**Soundness:** 2
**Presentation:** 3
**Contribution:** 3
**Rating:** 5
**Confidence:** 3

**Summary:**

This work presents a representation learning for image deformations by learning transformation groups in the latent space, and innovates the preservation of topology of object in the decoder by incorporating a smooth group composition module. Experiments on various databases along with the comparison with SOTA models are given.

**Strengths:**

The work proses a novel representation learning to preserve object topology in deformation. The whole paper is well-written.

**Weaknesses:**

1.	The descriptions and proofs of the given formula (2), (5), (6) are insufficient. Readers cannot fully understand the reason of the design.
2.	The qualitative experiments results are incomplete. No figures show the results on 3D Brains.

**Questions:**

1.	Please explain and verify the key formula proposed by this work. And explain the originality of the formula. For example, in formula (2), what is the second part? What is the relation to LDDMM?
2.	Please give more qualitative results, especially for 3D Brains database.

---

> ### Author Response · Authors · 2024-11-22
> **Response to Reviewer 2hVs**
>
> We thank the reviewers for their comments. Please find our point-to-point responses below.
>
> > *Comment 1.* The descriptions and proofs of the given formula (2), (5), (6) are insufficient. Readers cannot fully understand the reason of the design.
>
> The Eq. (2) in our background section has been extensively used in deformation estimation in groupwise images [1-4]. Given a template image, $I$, the objective of Eq. (2) is to find the geometric transformations between $I$ and each individual image, $I_n$, by optimizing two terms: (i) an image matching term that minimizes the error between deformed template image by the predicted transformations and the target individual image $I_n$, and (ii) a regularization term on the transformation fields, parameterized by velocity $v$, to ensure the smoothness of the transformation fields. Following these similar principles, the Eq. (5) is a network loss function built on top of Eq. (2), with the difference that the transformation/velocity fields are predicted by the neural networks.
>
> Eq. (6) is an image classification loss (e.g., cross-entropy loss), with the difference in taking integrated geometric features (velocity) and image features (image intensities/textures). It has been shown that integrated geometric and image features can substantially improve the classification accuracy **[Fig. 1/7/9]**.
>
> > *Comment 2.* The qualitative experiments results are incomplete. No figures show the results on 3D Brains.
>
> We have added qualitative results for real 3D Brain MRIs in the revised version. Our findings show that ToRL demonstrates superior class-wise clustering, indicating stronger differentiation between Disease and Healthy classes **(Fig. 5)**. Furthermore, it exhibits enhanced topological consistency and smoother deformations from the reference image to individual images **(Fig. 6)**.
>
>
> **References**
>
> [1] Wang, Jian, and Miaomiao Zhang. "Geo-sic: learning deformable geometric shapes in deep image classifiers." Advances in Neural Information Processing Systems 35 (2022): 27994-28007.
>
> [2] Balakrishnan, Guha, et al. "An unsupervised learning model for deformable medical image registration." Proceedings of the IEEE conference on computer vision and pattern recognition. 2018.
>
> [3] De Vos, Bob D., et al. "A deep learning framework for unsupervised affine and deformable image registration." Medical image analysis 52 (2019): 128-143.
>
> [4] Arar, Moab, et al. "Unsupervised multi-modal image registration via geometry preserving image-to-image translation." Proceedings of the IEEE/CVF conference on computer vision and pattern recognition. 2020.

---

> > ### Author Response · Authors · 2024-11-25
> > **Looking forward to your post-rebuttal feedback!**
> >
> > Thank you again for the insightful comments and suggestions! Given the limited time remaining, we eagerly anticipate your subsequent feedback. It would be our pleasure to offer more responses to further demonstrate the effectiveness of our methodology.
> >
> > In our previous response, we have thoroughly reviewed your comments and provided responses summarized as follows:
> >
> > - Explained the given formula 2, 5, and 6 and their originality.
> > - Provided qualitative results for 3D brain datasets
> >
> >
> > We hope that the additional formula description and qualitative visualization has convinced you of the merits of this paper. Please feel free to let us know if there are additional questions.
> >
> > We want to thank you again for finding our proposed representation learning novel. Additionally, we wish to express our gratitude once again to you for your insightful feedback. Incorporating your suggestions has undoubtedly enhanced the clarity of our work.
> >
> > We deeply appreciate your time and effort!
> >
> > Best regards, ToRL Authors

---

> > ### Author Response · Authors · 2024-11-26
> > **Seeking further discussions and post-rebuttal feedback**
> >
> > Dear Reviewers,
> >
> > We would like to thank you once again for dedicating your time to review our paper.
> >
> > We understand that your schedule may be quite busy, especially during the holidays. As the discussion period is expected to conclude in a few days, we look forward to hearing your feedback regarding whether we have satisfactorily addressed your concerns in the equation explanations and qualitative visualizations. Please let us know if we can help you with any information/experiments/explanations. Your insights will be highly valued and deeply appreciated. Thanks.
> >
> >
> > Best regards, ToRL Authors

---

> > > ### Author Response · Authors · 2024-12-02
> > > **Request for Final Feedback**
> > >
> > > Dear Reviewer 2hVs,
> > >
> > > We would like to thank you once again for dedicating your time to review our paper. As the discussion period is expected to conclude shortly, we look forward to hearing your feedback regarding whether we have satisfactorily addressed your concerns in our Additional Clarifications. If so, we kindly request a reevaluation of the scores to positively reflect these adjustments.
> > >
> > > Please let us know if we can help you with any other information or queries. Your insights will be highly valued and deeply appreciated.
> > >
> > > Best regards, ToRL Authors

---

> > > > ### Comment · Reviewer_2hVs · 2024-12-03
> > > > **keep original score**
> > > >
> > > > Thank  you for your answers to the questions and I am sorry for the delay of the response. Looking at the explaination of the terms in formula, there is still no discussion on the relation to LDDMM. Actually, I couldn't find the novelty in the formula and design. Although you would add figures on 3D brains, the original version was incomplete and not well-prepared. I'll keep the original scores.

---

> > > > > ### Author Response · Authors · 2024-12-03
> > > > > **Further Clarification**
> > > > >
> > > > > Thanks, Reviewer 2hVs for your reply and comments.
> > > > >
> > > > > > Novelty in the formula and design: Actually, I couldn't find the novelty in the formula and design.
> > > > >
> > > > > Please allow us to clarify that we specifically stated our contribution in line 89-95, where we mentioned that **ToRL for the first time** *(i) modeled complex objects' topology via learned transformation groups in the latent space of object deformations derived from images (**TGM, Latent representation of transformation groups**, L174-L225)*, (ii) *incorporated an integrated smooth group composition module in the deformation space to ensure the preservation of topology-
> > > > > ical properties throughout the learning process (**Topology-preserving decoder with group composition module**, L237-L257)*, and (iii) *eliminated the impractical requirement of having a reference image to predict object deformations during inference*.
> > > > >
> > > > > Regarding the formulation novelty, Eq. (4) represents our novel group composition module and Eq. (5) details the novel transformation group module through $\phi_n^c(v_n^c(z^c_{N_c}(\Uptheta))$ (L186-L187), which generates $\phi_n^c$ from learned $v_n^c$ at each layer of the decoder with a smoother regularization $\| \nabla v_n^c(\Uptheta)\|$. Please refer to the L258-L268. While existing networks only generate the transformation from the predicted velocity from the UNet, we learned these transformations through the newly introduced TGM module at each layer of the newly designed decoder structured upon the group composition module.
> > > > >
> > > > > > There is still no discussion on the relation to LDDMM
> > > > >
> > > > > We utilized a stationary velocity field (SVF) to develop the transformation group module (TGM). While both SVF and LDDMM frameworks are being used for modeling smooth, invertible transformations, SVF uses a constant velocity field for simplicity and computational efficiency and LDDMM employs time-varying velocity fields for greater flexibility and precision in capturing complex deformations. Please allow us to clarify that one can easily replace the SVF module with LDDMM (Geo-SIC). Please note that ToRL (SVF-based framework) outperforms Geo-SIC (LDDMM-based) across all datasets.
> > > > >
> > > > > > Although you would add figures on 3D brains, the original version was incomplete and not well-prepared
> > > > >
> > > > > Thanks for your comment. While we acknowledge we couldn't provide the 3D brain figures because of the page limitations, we provided pictorial depictions for the other datasets and included extended analysis on the revised version. The new 3D brain visualization holds the previous findings which show our newly designed decoder (ToRL-ND) visualizes superior topological consistency and smoother deformations from the reference image to each image. However, we provided quantitative evaluation on all dataset in the original version. We would kindly request the reviewer to reconsider the additional 3D brain figures.
> > > > >
> > > > > While we respect your standings and are grateful for your findings, we kindly request to reconsider the contribution of our model (recognized by Reviewer W891 and tDX8, where W891 increased the initial points) and our experimental validation comprehensive (recognized by all the other Reviewers W891, Wezs, tDX8).

---

### Author Response · Authors · 2024-11-22
**Author Rebuttal: General Response**

We sincerely thank all reviewers' valuable comments, insightful suggestions, and valuable feedback. We are encouraged that the reviewers recognize the

- **model/technical contribution novel and well-formulated** *(2hVs, W891, tDX8)*
- **importance of eliminating the reference image requirement** *(W891, tDX8)*
- **experimental validation and ablation study comprehensive** *(W891, Wezs, tDX8)*.

Besides, reviewers **appreciate our novel decoder design** for ensuring structural continuity *(tDX8)* and **mathematical foundation clear** *(W891)*, and **overall approach inspiring** *(Wezs)*. This helped improve our submission and strengthen our claims.

We have carefully addressed all the reviewers' concerns and hope the improvements and clarifications will be considered. Their feedback has helped improve our manuscript, and the changes are summarized below.

- Added qualitative visualizations on brain dataset **[2hVs]** *(Fig. 6)*

- Performed experiments on diverse real-world 2D facial expression datasets having diverse image qualities. **[Wezs, W891, tDX8]** *(Tab. 1/2/3/4; Fig. 5)*

- Reported computation costs of all models, including parameter counts, computational load (FLOPS), training/testing times, and their trade-offs with performance accuracy. **[Wezs, tDX8]** *(Tab. 5)*

- Validated individual components of ToRL. **[W891]** *(Appendix Fig. 8/9)*

- Evaluated additional metrics to assess the necessity of topology preservation. **[W891]** *(Tab. 4)*

- Performed experiments on the model’s robustness to input perturbations. **[W891]**  *(Fig. 8)*


For clarity, we highlight the revised part of the manuscript in blue color.

---

### Meta-Review · Area_Chair_NMaF · 2024-12-19

**Metareview:**

This submission introduces a topology-preserving representation learning framework that captures object deformations from images by explicitly modeling transformation groups in the latent feature space. The method incorporates a smooth group composition module in the decoder to preserve topological properties and thus removes dependency on reference images during inference. Experimental work investigates 2D shape and 3D medical imaging datasets, demonstrating improved performance in classification tasks and topology preservation in comparison with existing approaches.

This paper was discussed at length with the SAC. After reviewing the paper, rebuttal and resulting discussions the consensus was that although the review process has helped to strengthen the work, open reviewer concerns relating to experiments, writing, methodology can likely be better addressed with deeper revisions that will benefit from an additional round of review. AC therefore recommends rejection on this occasion however authors are encouraged to take onboard feedback for re-submission to an alternative future venue.

**Additional Comments On Reviewer Discussion:**

The paper received four reviews resulting in: two borderline accepts and two borderline rejects.

Reviewers noted positive aspects relating to the technical contribution, the utility of removing the reference image and the experimental work. Negative reviewer comments initially raised concerns relating to the mathematical exposition, qualitative experimental results, credit assignment of individual components, missing analyses involving topological metrics and model robustness, lack of experimental work considering real-world diffeomorphism settings and additional downstream tasks, poor writing and structure, factual errors and unsupported claims, and computational cost.

The author rebuttal could resolve a subset of concerns through discussion and presentation of new results in the updated manuscript. Post-rebuttal two reviewers remain explicitly unconvinced citing (still) open issues pertaining to experiments, writing, methodology and the preliminary state of the manuscript.

---

### Decision · Program_Chairs · 2025-01-22

Reject